# Learning with Noisy Correspondence
# for Cross-modal Matching

**Zhenyu Huang**[*]
College of Computer Science
Sichuan University, China
zyhuang.gm@gmail.com

**Guocheng Niu**
Baidu Inc., China
niuguocheng@baidu.com

**Xiao Liu**
TAL Education Group
liuxiao15@tal.com

**Wenbiao Ding**
TAL Education Group
dingwenbiao@tal.com

**Xinyan Xiao**
Baidu Inc., China
xiaoxinyan@baidu.com

**Hua Wu**
Baidu Inc., China
wu_hua@baidu.com

**Xi Peng**[†]
College of Computer Science
Sichuan University, China
pengx.gm@gmail.com

## Abstract

Cross-modal matching, which aims to establish the correspondence between two different modalities, is fundamental to a variety of tasks such as cross-modal retrieval and vision-and-language understanding. Although a huge number of cross-modal matching methods have been proposed and achieved remarkable progress in recent years, almost all of these methods implicitly assume that the multimodal training data are correctly aligned. In practice, however, such an assumption is extremely expensive even impossible to satisfy. Based on this observation, we reveal and study a latent and challenging direction in cross-modal matching, named noisy correspondence, which could be regarded as a new paradigm of noisy labels. Different from the traditional noisy labels which mainly refer to the errors in category labels, our noisy correspondence refers to the mismatch paired samples. To solve this new problem, we propose a novel method for learning with noisy correspondence, named Noisy Correspondence Rectifier (NCR). In brief, NCR divides the data into clean and noisy partitions based on the memorization effect of neural networks and then rectifies the correspondence via an adaptive prediction model in a co-teaching manner. To verify the effectiveness of our method, we conduct experiments by using the image-text matching as a showcase. Extensive experiments on Flickr30K, MS-COCO, and Conceptual Captions verify the effectiveness of our method. The code could be accessed from www.pengxi. me.

## 1 Introduction

As one of the most fundamental techniques in multimodal learning, cross-modal matching aims to bridge different modalities. In recent years, some cross-modal matching methods [19, 11, 7, 26] have been proposed based on Deep Neural Networks (DNNs), which achieved remarkable progress in a

---

[*]Some parts of the work was done while Zhenyu Huang was an internship at Baidu Inc.
[†]Corresponding author.

35th Conference on Neural Information Processing Systems (NeurIPS 2021).

variety of applications, such as clustering [29, 24], image/video captioning [1, 44, 22], cross-modal retrieval [40, 19, 13], and visual question answering [9].

In general, most existing cross-modal matching methods embed different modalities into a common space wherein the similarity of positive cross-modal pairs is maximized and that of the negative ones is minimized. Although these methods have achieved promising results, their success depends on an implicit data assumption, *i.e.*, the training data are correctly aligned across modalities. For example, in the vision-and-language tasks, the text needs to accurately describe the image content, and vice versa. In practice, however, it is extremely expensive and time-consuming to annotate or collect such data pairs. Especially, considering the data collected from the Internet [35, 14], it is inevitable to collect some mismatched pairs which are wrongly treated as the matched ones. To the best of our knowledge, such a special noisy label (correspondence) problem has been ignored so far, which will remarkably degrade the performance of matching methods as shown in our experiments.

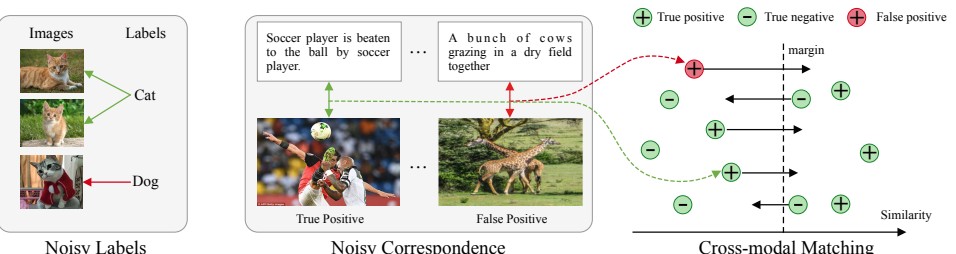

Figure 1: Noisy labels vs. Noisy Correspondence. We denote the noisy samples with red lines and clean samples with green lines. The traditional noisy labels mainly refer to the errors in category labels, while the noisy correspondence refers to the alignment errors in paired data. For the noisy correspondence in cross-modal matching, the true positive pair correctly guides the cross-modal matching, while the false positive pair causes incorrect supervision for training.

Based on the above observation, we reveal a new paradigm for the noisy labels, named noisy correspondence. Different from the traditional noisy labels, the noisy correspondence refers to the alignment errors in paired data rather than the errors in category annotations (see Fig. 1). To the best of our knowledge, there is no effort has been devoted to study this new problem and the closest paradigm might be the partially view-aligned problem (PVP) [12, 41]. However, PVP is remarkably different from noisy correspondence, and the latter is more practical than the former. To be specific, PVP focuses on that the cross-modal alignment is totally unavailable, whereas the noisy correspondence focuses on that some correspondences are incorrect. In addition, PVP assumes that some correctly aligned data are available for training, whereas our noisy correspondence assumes that the clean and noisy data are mixed.

To solve the noisy correspondence problem in cross-modal matching, we propose a novel method, named Noisy Correspondence Rectifier (NCR). Our method is based on the memorization effect of DNNs observed in [3, 39], *i.e.*, DNNs tend to learn the simple patterns before fitting noisy samples. Motivated by this empirical observation, NCR divides the data into two relative accurate data partitions, *i.e.*, "noisy" and "clean" subsets, based on their loss difference. After that, NCR employs an adaptive prediction function for label rectifying so that the false positives and the true positives could be identified from the "clean" and the "noisy" subsets, respectively. Furthermore, we propose a novel triplet loss for robust cross-modal matching by recasting the rectified labels as the soft margin.

The main contributions and novelties of this paper could be summarized as below. i) We reveal a new problem in cross-modal analysis, which is also a new paradigm for noisy labels, termed noisy correspondence. Different from the traditional noisy labels, the noisy correspondence refers to the alignment errors in paired data instead of the errors in category annotations. To the best of our knowledge, this work could be the first study on this problem. ii) To solve the noisy correspondence problem, we propose a novel method for learning with noisy correspondence, named Noisy Correspondence Rectifier (NCR). One major novelty of NCR is that the rectified label is elegantly recasted as the soft margin of a triplet loss so that the robust cross-modal matching could be achieved. iii) To verify the effectiveness of our method, we conduct experiments on the image-text matching task. Extensive experiments on three challenging datasets verify the effectiveness of our method in the synthesized and real noises.

# 2 Related works

In this section, we briefly introduce some recent developments in cross-modal matching and learning with noisy labels.

## 2.1 Cross-modal Matching

Most existing cross-modal matching works seek to learn a common space wherein different modalities are comparable. In general, existing works could be roughly divided into two categories: 1) Coarse-grained Matching. It often utilizes multiple neural networks to compute a global feature and each network is used for a specific modality [17, 37, 8]. For example, Kiros et al. [17] use a Convolutional Neural Network (CNN) and a Gated Recurrent Unit (GRU) to obtain the image and text features, while enforcing the similarity of positive pairs larger than that of the negative ones. To further boost the matching performance, VSE++ [8] uses some representative negatives to improve the discrimination of the model. 2) Fine-grained Matching. It seeks to measure the fine-grained similarity for cross-modal matching [19, 21, 7]. For example, SCAN [19] proposes learning the latent semantic correspondence between the image regions and words that are extracted by bottom-up attention [1] and GRU, respectively. VSRN [21] adopts a graph convolutional network for semantic reasoning. SGRAF [7] proposes constructing a similarity graph to reason the similarity and adopting an attention filtration technique to eliminate the less-meaningful alignments. Recently Chun et al. [6] introduce a new paradigm for cross-modal matching, i.e. possible many-to-many correspondence that existed in the image and captions. To achieve this, they propose to use probabilistic representations to model the possible one-to-many correspondence.

Although promising results have been achieved in recent years, the existing methods heavily rely on the correctly aligned data. In practice, however, such well-matched data is expensive and time-consuming to collect. Moreover, some recent works [35, 14] show that a large-scale dataset collected from the wild could remarkably improve the performance of the model. However, such a data will inevitably contain some mismatched pairs. Hence, it is highly expected to develop some methods which are robust against the noisy correspondence, which has not been studied as far as we know. Different from the many-to-many correspondence [6] between image and captions, NCR reveals the noisy correspondence problem which refers to the alignment errors of image-text pairs and proposes to eliminate the negative impact from noisy pairs for downstream tasks.

## 2.2 Learning with Noisy Labels

To handle the possible noisy annotations in the training data, a large number of methods have been proposed and almost all of them focus on the classification task [36, 27]. To reduce the negative impact of the noisy labels, the existing works often resort to robust architecture design, regularization, loss adjustment, or sample selection methods. Here, we mainly introduce the last two approaches which are most related to this work. To be specific, the loss adjustment achieves robustness by adjusting the contribution of clean and noisy samples w.r.t. the loss. For example, Reed et al. [32] proposed a bootstrapping loss based on the model predictions for loss correction. Zhang et al. [45] provided some theoretical explanations for the label correction along with a new label correction algorithm. Different from the loss adjustment methods, sample selection methods aim to select clean samples from a noisy dataset. For example, Arpit et al. [3] showed that DNNs tend to learn simple patterns before fitting noisy samples, namely the memorization effect. Motivated from this, Arazo et al. [2] proposed treating the samples with small loss as the clean samples. To avoid the selection bias of clean samples, Co-teaching methods [10, 43] use the samples with small loss to iteratively train two networks. In recent, DivideMix [20] adopts the MixMatch method [4] for semi-supervised learning with the clean and noisy samples.

Unlike the above noisy label studies, this paper focuses on the noisy correspondence problem which considers mismatched multimodal data pairs instead of incorrectly annotated data points. Besides the difference in the problem, this work is also different from the aforementioned studies in the methodology. To be specific, in cross-modal matching, it is impossible to directly adopt these noise label learning methods to solve the noisy correspondence problem due to the following two reasons. First, most of the noisy label learning methods propose to use the model's prediction for label rectifying in the scenario of classification, while it is intractable to directly predict the correspondence of given pairs in matching models. Second, even if we can rectify the noisy correspondence somehow,

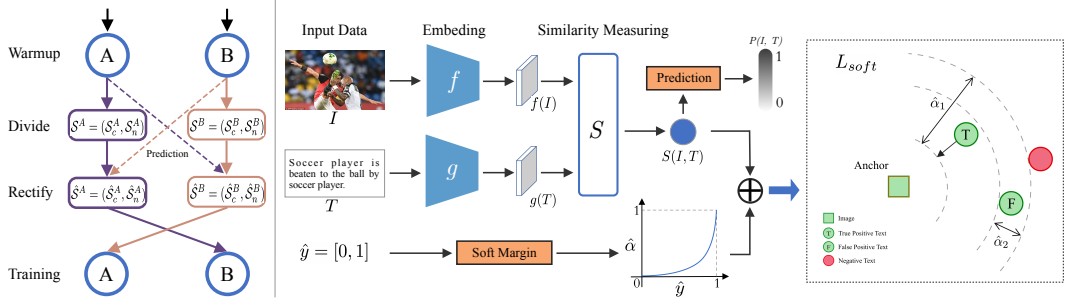

| | |
|---|---|
| (a) Training pipeline | (b) Robust image-text matching network ($A$ and $B$) |

Figure 2: Overview of the proposed method. (a) Training pipeline of NCR. NCR consists of two individual networks (A, B) which work in the manner of co-teaching. In brief, NCR first warmup the networks (A, B) on the original training data using the loss $L_w$ which is also used for per-sample loss computation. Then, based on the memorization effect of DNNs, NCR divides the training data into clean and noisy subsets at each epoch using either A or B, *i.e.*, $\mathcal{S}^A = (\mathcal{S}_c^A, \mathcal{S}_n^A)$ and $\mathcal{S}^B = (\mathcal{S}_c^B, \mathcal{S}_n^B)$. After that, NCR will co-rectify the correspondence of $\{\mathcal{S}^A, \mathcal{S}^B\}$ and obtain $\{\hat{\mathcal{S}}^A, \hat{\mathcal{S}}^B\}$ using an adaptive prediction function. Finally, $\hat{\mathcal{S}}^A$ and $\hat{\mathcal{S}}^B$ will be used to train the network $B$ and $A$ in a swapping way. (b) Robust image-text matching network. For example, the network A projects the image and text by the modal-specific networks $f$ and $g$, respectively. Then the similarity $S(I, T)$ is computed on the extracted features $f(I)$ and $g(T)$. To achieve robust image-text matching, the rectified soft labels are recast as the soft margin of our loss $L_{soft}$. As shown, for a given anchor, $L_{soft}$ will enforce the true positive to closer to it than the negative by a large margin $\hat{\alpha}_1$, and meanwhile the false positive will has a small margin $\hat{\alpha}_2$.

the rectified real-valued labels are incompatible with the existing matching methods since most of them assume the given labels are binary. To address these problems, NCR proposes an adaptive prediction function and a novel triplet loss by recasting the soft labels as soft margins.

## 3    The Proposed Method

In this section, we elaborate on the proposed method, *i.e.*, Noisy Correspondence Rectifier (NCR) which could be the first work to solve the noisy correspondence problem in cross-modal matching. In Section 3.1, we introduce the co-divide module which splits the training data into the clean and noisy subsets. After that, we introduce how to rectify the labels with an adaptive prediction function in Section 3.2. Finally, we detail how to combine the co-divide and co-rectify modules to achieve robust cross-modal matching in Section 3.3.

### 3.1    Co-divide

Without loss of generality, we first introduce the cross-modal matching task by taking the image-text matching as a showcase. Given the training data $\mathcal{D} = \{(I_i, T_i, y_i)\}_{i=1}^N$, where $N$ is the data size, $(I_i, T_i)$ is an image-text pair and $y_i \in \{1, 0\}$ indicates that the pair belong to the same instance (positive) or not (negative). For the noisy correspondence case, it defines that an unknown portion of $\mathcal{D}$ is mismatched, *i.e.*, $(I_i, T_i)$ is a negative pair but wrongly labeled as $y_i = 1$. To solve such a noisy correspondence problem, we propose NCR to achieve robust cross-modal matching.

To begin, we project the visual and textual modalities into a shared space via two modal-specific networks $f$ and $g$, respectively. Then the similarity of the given image-text pairs is computed through $S(f(I), g(T))$. For simplicity, we denote $S(f(I), g(T))$ as $S(I, T)$ in the following. Some early empirical studies [3] show that DNNs tend to first learn simple samples and then gradually fit the noisy samples. This so-called memorization effect of DNNs will lead to a relatively low loss for the clean samples. Motivated by this, we utilize the difference of loss distribution between the clean samples and noisy samples to divide the training data like [10, 43, 2, 20]. Specifically, given a matching model $(f, g, S)$, we compute the per-sample loss through:

$$\ell_{(f,g,S)} = \{\ell_i\}_{i=1}^N = \{L_w(I_i, T_i)\}_{i=1}^N \tag{1}$$

where $L_w$ is defined as:

$$L_w(I,T) = \sum_{\hat{T}}[\alpha - S(I,T) + S(I,\hat{T})]_+ + \sum_{\hat{I}}[\alpha - S(I,T) + S(\hat{I},T)]_+, \qquad (2)$$

where $(I,T)$ is a positive pair, $\alpha > 0$ denotes a given margin, and $[x]_+ = max(x,0)$. In the loss, the first term treats $I$ as queries taking over all negative text $\hat{T}$, while the second term treats $T$ as queries taking over all negative images $\hat{I}$. Then, we fit the per-sample loss of all training data by using a two-component Gaussian Mixture Model [20, 30]:

$$p(\ell|\theta) = \sum_{k=1}^{K} \beta_k \phi(\ell|k), \qquad (3)$$

where $\beta_k$ and $\phi(\ell|k)$ are the mixture coefficient and the probability density of the $k$-th component, respectively. Based on the memorization effect of DNNs, we treat the component with a lower mean value (*i.e.*, lower loss) as the clean set, and the other as the noisy set. To optimize the GMM, we adopt the Expectation-Maximization algorithm. Moreover, we compute the posterior probability $w_i = p(k|\ell_i) = p(k)p(\ell_i|k)/p(\ell_i)$ as the clean probability of $i$-th sample, where $k$ is the Gaussian component with the lower mean. By setting a threshold to $\{w_i\}_{i=1}^N$, we divide the data into clean and noisy subsets. For simplicity, we set the threshold to $0.5$ through all experiments.

As observed in [10], it probably introduces error accumulation if the neural network is trained in a self-divide manner. To avoid such a situation, we adopt the co-teaching paradigm. Specifically, we individually train two networks $A = \{f^A, g^A, S^A\}$ and $B = \{f^B, g^B, S^B\}$ with different initializations and batch sequences. At each epoch, the network $A$ or $B$ will model its per-sample loss distribution with a GMM and divide the dataset into clean and noisy subsets which are then used for training the other network, *i.e.*, co-divide. Note that, before co-divide, a warmup process is conducted on all training data to achieve initial convergence with $L_w$ as defined in Eq. 2.

## 3.2 Co-Rectify

For either of A and B, the data $\mathcal{D}$ will be divided into the clean subset $\mathcal{S}_c = \{(I_i^c, T_i^c, y_i^c, w_i)\}_{i=1}^{N_c}$ and noisy subset $\mathcal{S}_n = \{I_i^n, T_i^n\}_{i=1}^{N_n}$. Then, the co-rectify module will correct the labels to recall the possible true positives from $\mathcal{S}_n$ and eliminate the negative impact of the possible false positives in $\mathcal{S}_c$. Formally, the network $k$ ($k \in \{A, B\}$) will rectify the labels of $\{\mathcal{S}_c, \mathcal{S}_n\}$ into $\{\hat{\mathcal{S}}_c, \hat{\mathcal{S}}_n\}$ for training itself. The rectified labels are determined by:

$$\begin{cases} \hat{y}_i^c = w_i y_i^c + (1-w_i)P^k(I_i^c, T_i^c), & \forall(I_i^c, T_i^c, y_i^c, w_i^c) \in \mathcal{S}_c \\ \hat{y}_i^n = (P^A(I_i^n, T_i^n) + P^B(I_i^n, T_i^n))/2, & \forall(I_i^n, T_i^n) \in \mathcal{S}_n \end{cases} \qquad (4)$$

where $P^A(I,T)/P^B(I,T)$ denotes the predictions given by the network $A/B$. The roles of Eq. 4 are as below. On the one hand, as most pairs of $\mathcal{S}_c$ are true positive, Eq. 4 will use the original labels $y_i^c$ together with the model's prediction $P(I_i^c, T_i^c)$ to rectify the correspondence. On the other hand, as most pairs of $\mathcal{S}_n$ are false positive, Eq. 4 will discard the original labels and rectify the labels by averaging the predictions $P(I_i^n, T_i^n)$ from the networks A and B.

Another key contribution of Eq. 4 is designing the prediction function $P(I,T)$ that could accurately predict whether the given pairs are positive or negative. Unlike the tasks like classification, image-text matching aims at computing the similarity rather than predicting the label of given image-text pairs. To this end, a straightforward approach is to predict the pairs by setting a threshold on the similarity. However, such a method requires to specify the threshold value, which is a daunting task because the optimal value is actually the similarity boundary of positive and negative pairs and hard to be manually specified. Alternatively, the following adaptive prediction function $P(I,T)$ is proposed, which could work in a data-driven way,

$$P(I,T) = \Theta(s)/\tau$$
$$s = S(I,T) - (\frac{1}{b}\sum_{\hat{T}} S(I,\hat{T}) + \frac{1}{b}\sum_{\hat{I}} S(\hat{I},T))/2, \qquad (5)$$

where $b$ is the batch size, $\Theta(\cdot)$ clamps the elements into the range of $[0,\alpha]$, $s$ is the similarity margin between the given pair $(I,T)$ to the mean of the negatives in a mini-batch, $\tau$ is the average similarity

margin of the largest $10\%$ pairs in terms of $s$. This implies that the data have at least $10\%$ clean pairs which could be regarded as a similarity anchor for prediction. Intuitively, the pairs with the similarity margin larger than $\tau$ would be predicted as 1, otherwise $[0, 1)$.

---

**Algorithm 1:** Noisy Correspondence Rectifier

---

**Input:** A given training data $\mathcal{D}$, matching models $A = (f^A, g^A, S^A)$ and $B = (f^B, g^B, S^B)$

1   Warmup the model $(A, B)$ using $L_w$.
2   **for** *n=1:num_epoch* **do**
3      $\mathcal{W}^A = \{w_i^A\}_{i=0}^N \leftarrow GMM(\mathcal{D}, B)$
4      $\mathcal{W}^B = \{w_i^B\}_{i=0}^N \leftarrow GMM(\mathcal{D}, A)$
5      **for** *k={A, B}* **do**
6          $\mathcal{S}_c^k = \{(I_i, T_i, y_i, w_i)|w_i \geq 0.5, \forall (I_i, T_i, y_i, w_i) \in (\mathcal{D}, \mathcal{W}^k)\}$
7          $\mathcal{S}_n^k = \{(I_i, T_i)|w_i < 0.5, \forall (I_i, T_i) \in (\mathcal{D}, \mathcal{W}^k)\}$
8          **for** *j=num_steps* **do**
9             Sample a mini-batch $(\mathcal{B}_j^c, \mathcal{B}_j^n)$ from $(\mathcal{S}_c^k, \mathcal{S}_n^k)$;
10             Rectify the labels of $(\mathcal{B}_j^c, \mathcal{B}_j^n)$ into $(\hat{\mathcal{B}}_j^c, \hat{\mathcal{B}}_j^n)$ using Eq. 4–5;
11             Train the network $k$ on $(\hat{\mathcal{B}}_j^c, \hat{\mathcal{B}}_j^n)$ by optimizing $L_{soft}$.

**Result:** Matching models $(A, B)$

---

### 3.3 Robust Cross-modal Matching

Exiting cross-modal matching methods can only handle the binary labels which are incompatible with the soft labels rectified by NCR. To achieve robust image-text matching, we propose a novel triplet loss $L_{soft}$ by recasting the rectified labels as the soft margin. Mathematically,

$$L_{soft}(I_i, T_i) = [\hat{\alpha}_i - S(I_i, T_i) + S(I_i, \hat{T}_h)]_+ + [\hat{\alpha}_i - S(I_i, T_i) + S(\hat{I}_h, T_i)]_+, \tag{6}$$

where $\hat{I}_h = \text{argmax}_{I_j \neq I_i} S(I_j, T_i)$ and $\hat{T}_h = \text{argmax}_{T_j \neq T_i} S(I_i, T_j)$ are the most similar negatives in the mini-batch for a given positive pair $(I_i, T_i)$ similar to VSE++ [8]. The soft margin $\hat{\alpha}_i$ is adaptively determined by:

$$\hat{\alpha}_i = \frac{m^{\hat{y}_i} - 1}{m - 1}\alpha, \tag{7}$$

where $m$ is the curve parameter, and $\hat{y}_i$ is the rectified label. The above formulation is designed to achieve the following goal, *i.e.*, $\hat{\alpha}_i$ will be assigned a small value if $\hat{y}_i$ is close to 0, and a large value otherwise. Thanks to Eq. 6–7, the similarity of the pair $(I, T)$ will be larger than that of the negatives by an adaptive margin $\hat{\alpha}_i$.

Despite the adaptive margin, another major difference between $L_w$ and $L_{soft}$ is that $L_{soft}$ will use the hard negatives which are the most similar negative pairs. Although the hard negatives are helpful in improving the performance, $L_w$ cannot be beneficial from it due to the existence of noisy correspondence. Specifically, it is expected that only the similarity of the true positives is larger than that of the hard negatives. However, in the case of noisy correspondence, the similarity of the false positives will also be larger than that of the hard negatives, thus leading to the unavailability of the hard negatives for $L_w$ during the co-divide stage. The detail of NCR is presented in Algorithm. 1.

### 3.4 Discussions on Matching Loss

To achieve robust cross-modal matching with the refined soft labels, we design a soft Triplet loss by recasting the labels into soft margins. Recently, there are some works have been proposed to handle the soft labels in the matching model. For example, Wray et al. [38] recast the soft similarity into binary labels by directly setting a threshold on the predicted similarity. Kim et al. [15] proposes a log-ratio matching loss with a regularization defined by the label distance ratio, which is computed by the continuous labels. Liu et al. [25] introduces the hubness problem in image-text matching and proposes to consider all samples in a mini-batch and weights them according to both local and global statistics. Wray et al. [38] recasts the soft similarity into binary labels by directly setting a threshold on the predicted similarity. Different from them, NCR proposes to recast the rectified soft labels into

soft margins in the triplet loss by assigning large margins to the true positive pairs and small ones to the false positive pairs. As a result and more importantly, our loss is specifically designed to solve the noisy correspondence problem whereas existing ones are not.

# 4 Experiment

In this section, we carry out experiments to verify the effectiveness of NCR in robust image-text matching. In the experiments, we use three benchmark datasets including Flickr30K [42], MS-COCO [23], and Conceptual Captions [35]. Among them, Conceptual Captions is with real noisy correspondence from the wild, and Flickr30K and MS-COCO are with simulated noisy correspondence.

## 4.1 Datasets and Performance Measurements

Three datasets are used in our experiments. To be specific, Flickr30K contains 31,000 images collected from the Flickr website with five captions each. Following [19], we use 1,000 images for validation, 1,000 images for testing, and the rest for training. MS-COCO contains 123,287 images with five captions each. We follow the data partition in [19] which consists of 113,287 training images, 5,000 validation images, and 5,000 test images. As Flickr30K and MS-COCO are well annotated, we simulate the noisy correspondence by randomly shuffling the captions of training images for a specific percentage, denoted by noise ratio. Conceptual Captions is a large-scale data consisting of 3.3M images with a single caption each. As this data set is harvested from the Internet, about $3\% \sim 20\%$ correspondences are incorrect [35]. In our experiments, we use a subset of Conceptual Captions for evaluation, named CC152K. Specifically, we randomly select 150,000 samples from the training split for training, 1,000 samples from the validation split for validation, and 1,000 samples from the validation split for testing.

Following [19], for all images, we take the Faster-RCNN [33] detector provided by [1] to extract the top 36 region proposals of which each is encoded as a 2048-dimensional feature. For evaluation, we take the recall at K (R@K) as the measurement. In short, R@K is the fraction of queries for which the correct item is retrieved in the closest K points to the query. In the experiments, we report R@1, R@5, and R@10 for a comprehensive evaluation.

## 4.2 Implementation Details

NCR is a general framework which could enable almost all existing cross-modal matching methods robust against noisy correspondence. To verify the effectiveness of our framework, SGR [7] is chosen to guarantee the robustness because it is the state of the art in image-text matching. In brief, the image regions and words are projected into a shared embedding space through a full-connected layer (*i.e.*, $f$) and a Bi-GRU [34] (*i.e.*, $g$), respectively. For the similarity function $S$, it will compute the similarity between the given image and text by combining the local and global features with the help of a graph reasoning technique proposed in [18]. Due to the space limitation, we leave more details and results in the supplemental material.

We train our network using the Adam optimizer [16] with the default parameters and a batch size of 128. For fair comparisons, the networks $f$ and $g$ are the same with SGR, *i.e.*, the word embedding size is 300 and the joint embedding space size is 2048. In addition, we fix the margin $\alpha = 0.2$ and $m = 10$ for the soft margin through the experiments. At the inference stage, we average the similarities predicted by network $A$ and $B$ for the retrieval evaluation. To avoid overfitting, we choose the best checkpoint in terms of the sum of the recalls on the validation set.

## 4.3 Comparisons with State of The Arts

In this section, we conduct comparisons on the three datasets. The baselines include SCAN [19], VSRN [21], IMRAM [5], SGRAF, SGR and SAF [7]. For Flickr30K and MS-COCO, we report the results with three different noise ratios, *i.e.*, 0%, 20%, and 50%. In addition, we also report the results of SGR on the clean Flickr30K and MS-COCO by discarding the noisy pairs, denoted by SGR-C. Clearly, SGR-C is a quite strong baseline since the used data does not contain noisy correspondence. We do not report the results of SGRAF and SAF on the clean datasets since our framework only extends SGR in this paper.

When the noise rate is $0\%$, we directly refer to the results reported in the corresponding papers. For the noisy cases, we train the baseline models with the recommended settings three times and report the best result. Note that for SGR, we found it is very sensitive to the noisy correspondence, as shown in Table. 2. To obtain a desirable result, we experimentally employ a pre-training process to SGR (denoted by SGR*), namely, training the model with the vanilla triplet loss without hard negatives and then following the standard pipeline of SGR.

Table 1: Image-Text Retrieval on Flickr30K and MS-COCO 1K.

| | | Flickr30K | | | | | | MS-COCO | | | | | |
| | | Image → Text | | | Text → Image | | | Image → Text | | | Text → Image | | |
| Noise | Methods | R@1 | R@5 | R@10 | R@1 | R@5 | R@10 | R@1 | R@5 | R@10 | R@1 | R@5 | R@10 |
|---|---|---|---|---|---|---|---|---|---|---|---|---|---|
| | SCAN | 67.4 | 90.3 | 95.8 | 48.6 | 77.7 | 85.2 | 69.2 | 93.6 | 97.6 | 56.0 | 86.5 | 93.5 |
| | VSRN | 71.3 | 90.6 | 96.0 | 54.7 | 81.8 | 88.2 | 76.2 | 94.8 | 98.2 | 62.8 | 89.7 | 95.1 |
| | IMRAM | 74.1 | 93.0 | 96.6 | 53.9 | 79.4 | 87.2 | 76.7 | 95.6 | 98.5 | 61.7 | 89.1 | 95.0 |
| 0% | SAF | 73.7 | 93.3 | 96.3 | 56.1 | 81.5 | 88.0 | 76.1 | 95.4 | 98.3 | 61.8 | 89.4 | 95.3 |
| | SGR | 75.2 | 93.3 | 96.6 | 56.2 | 81.0 | 86.5 | 78.0 | 95.8 | 98.2 | 61.4 | 89.3 | 95.4 |
| | SGRAF | **77.8** | **94.1** | 97.4 | 58.5 | 83.0 | 88.8 | **79.6** | **96.2** | **98.5** | 63.2 | **90.7** | **96.1** |
| | **NCR** | 77.3 | 94.0 | **97.5** | **59.6** | **84.4** | **89.9** | 78.7 | 95.8 | **98.5** | **63.3** | 90.4 | 95.8 |
| | SCAN | 59.1 | 83.4 | 90.4 | 36.6 | 67.0 | 77.5 | 66.2 | 91.0 | 96.4 | 45.0 | 80.2 | 89.3 |
| | VSRN | 58.1 | 82.6 | 89.3 | 40.7 | 68.7 | 78.2 | 25.1 | 59.0 | 74.8 | 17.6 | 49.0 | 64.1 |
| | IMRAM | 63.0 | 86.0 | 91.3 | 41.4 | 71.2 | 80.5 | 68.6 | 92.8 | 97.6 | 55.7 | 85.3 | 91.0 |
| 20% | SAF | 51.0 | 79.3 | 88.0 | 38.3 | 66.5 | 76.2 | 67.3 | 92.5 | 96.6 | 53.4 | 84.5 | 92.4 |
| | SGR* | 62.8 | 86.2 | 92.2 | 44.4 | 72.3 | 80.4 | 67.8 | 91.7 | 96.2 | 52.9 | 83.5 | 90.1 |
| | SGR-C | 72.8 | 90.8 | 95.4 | 56.4 | 82.1 | 88.6 | 75.4 | 95.2 | 97.9 | 60.1 | 88.5 | 94.8 |
| | **NCR** | **75.0** | **93.9** | **97.5** | **58.3** | **83.0** | **89.0** | **77.7** | **95.5** | **98.2** | **62.5** | **89.3** | **95.3** |
| | SCAN | 27.7 | 57.6 | 68.8 | 16.2 | 39.3 | 49.8 | 40.8 | 73.5 | 84.9 | 5.4 | 15.1 | 21.0 |
| | VSRN | 14.3 | 37.6 | 50.0 | 12.1 | 30.0 | 39.4 | 23.5 | 54.7 | 69.3 | 16.0 | 47.8 | 65.9 |
| | IMRAM | 9.1 | 26.6 | 38.2 | 2.7 | 8.4 | 12.7 | 21.3 | 60.2 | 75.9 | 22.3 | 52.8 | 64.3 |
| 50% | SAF | 30.3 | 63.6 | 75.4 | 27.9 | 53.7 | 65.1 | 30.4 | 67.8 | 82.3 | 33.5 | 69.0 | 82.8 |
| | SGR* | 36.9 | 68.1 | 80.2 | 29.3 | 56.2 | 67.0 | 60.6 | 87.4 | 93.6 | 46.0 | 74.2 | 79.0 |
| | SGR-C | 69.8 | 90.3 | 94.8 | 50.1 | 77.5 | 85.2 | 71.7 | 94.1 | 97.7 | 57.0 | 86.6 | 93.7 |
| | **NCR** | **72.9** | **93.0** | **96.3** | **54.3** | **79.8** | **86.5** | **74.6** | **94.6** | **97.8** | **59.1** | **87.8** | **94.5** |

**Results on Flickr30K & MS-COCO**. Table 1 shows the quantitative results on Flickr30K and MS-COCO. Note that for MS-COCO, we only report the results by averaging over 5 folds of 1K test images due to space limitation, and leave the results on the full 5K test images in the supplemental material. From the results, one could observe that NCR is competitive to SGRAF in the noise-free case, namely, NCR could achieve state-of-the-art performance even though it is proposed to achieve robustness. When the data is contaminated by the noisy correspondence, NCR remarkably outperforms all the baselines by a large margin. Even comparing with SGR-C which is trained on the clean data, NCR improves R@1 by $2.2\%$, $3.1\%$, $2.3\%$, and $2.9\%$ in these four valuations.

**Results on CC152K**. Table 2 shows the quantitative results on the CC152K which is with real noisy correspondences. From the results, one could see that our NCR consistently outperforms the evaluated models by a considerable margin in terms of all metrics. Specifically, NCR is $4.5\%$ and $5.4\%$ higher than the best baseline in terms of R@1 in text and image retrieval, respectively. Moreover, the large performance gap between SGR and SGR* shows the noise sensitivity of the original SGR.

Table 2: Image-Text Retrieval on CC152K.

| | Image → Text | | | Text → Image | | |
| Methods | R@1 | R@5 | R@10 | R@1 | R@5 | R@10 |
|---|---|---|---|---|---|---|
| SCAN (ECCV'18) | 30.5 | 55.3 | 65.3 | 26.9 | 53.0 | 64.7 |
| VSRN (ICCV'19) | 32.6 | 61.3 | 70.5 | 32.5 | 59.4 | 70.4 |
| IMRAM (CVPR'20) | 33.1 | 57.6 | 68.1 | 29.0 | 56.8 | 67.4 |
| SAF (AAAI'21) | 31.7 | 59.3 | 68.2 | 31.9 | 59.0 | 67.9 |
| SGR (AAAI'21) | 11.3 | 29.7 | 39.6 | 13.1 | 30.1 | 41.6 |
| SGR* (AAAI'21) | 35.0 | 63.4 | 73.3 | 34.9 | 63.0 | 72.8 |
| **NCR** | **39.5** | **64.5** | **73.5** | **40.3** | **64.6** | **73.2** |

## 4.4 Comparison to pre-trained model

In this section, we perform comparison to the large pre-trained model CLIP [31]. In brief, CLIP is trained on a massive dataset harvested from the Internet and thus presumably has a lot of noisy

image-text pairs. Such a comparison is helpful in understanding, big data based model (CLIP) or noisy correspondence modeling technique (NCR), which one is more favorable to handle the mismatching problem. More specifically, CLIP claims that using hundreds of million data could ignore the existence of possible noise, while we believe that a well-designed algorithm is essential to solve the noisy correspondence. Noticed, although some existing works including CLIP have slightly/indirectly realized the existence of the noise, NONE of them explicitly give a solution to solve this problem and explores the characteristics of the noise correspondence.

In the experiments, we conduct CLIP on the MS-COCO dataset under the following two settings, *i.e.*, Zero-shot and Fine-tune. In brief, the first setting directly uses the released pre-trained CLIP to perform inference on MS-COCO, and the second fine-tunes the pre-trained model using the noisy training data of MS-COCO. As CLIP only released some pre-trained models and inference code [3], we use the non-official code [4] to fine-tune the model with 32 epochs for the fine-tune setting. Note that CLIP (ViT-L/14†) is unreleased and we report the results from the original paper [31]. One could observe that, although CLIP utilizes 400 million image-text pairs for pre-training, its performance inevitably degenerates during fine-tuning. In contrast, NCR achieves the matching performance with the presence of noisy correspondence, indicating the necessity of algorithm design.

Table 3: Comparison with CLIP on MS-COCO 5K.

| Noise Ratio | Methods | Image → Text | | | Text → Image | | |
|---|---|---|---|---|---|---|---|
| | | R@1 | R@5 | R@10 | R@1 | R@5 | R@10 |
| 0%, Zero-Shot | CLIP (ViT-L/14†) | **58.4** | 81.5 | 88.1 | 37.8 | 62.4 | 72.2 |
| | CLIP (ViT-B/32) | 50.2 | 74.6 | 83.6 | 30.4 | 56.0 | 66.8 |
| | **NCR** | 58.2 | **84.2** | **91.5** | **41.7** | **71.0** | **81.3** |
| 20%, Fine-tune | CLIP (ViT-B/32) | 21.4 | 49.6 | 63.3 | 14.8 | 37.6 | 49.6 |
| | **NCR** | **56.9** | **83.6** | **91.0** | **40.6** | **69.8** | **80.1** |
| 50%, Fine-tune | CLIP (ViT-B/32) | 10.9 | 27.8 | 38.3 | 7.8 | 19.5 | 26.8 |
| | **NCR** | **53.1** | **80.7** | **88.5** | **37.9** | **66.6** | **77.8** |

## 4.5 Experimental Analysis

In this section, we first conduct experiments to show the robustness and generalizability of the proposed method. Then, we investigate the effect of co-divide and co-rectify with the visualization results. After that, we carry out the ablation study to verify different components of NCR. Finally, we visually demonstrate some noisy cases detected by NCR.

### 4.5.1 Study on Robustness and Generalizability

To show the robustness of NCR, we conduct experiments on Flickr30K by increasing the noise ratio from 0% to 60% with an interval of 10%. In addition, to verify the generalizability of NCR to other image-text matching methods, we extend SCAN [19] by NCR, denoted by NCR-SCAN. Fig. 4 shows that NCR and NCR-SCAN perform more stable than SGR and SCAN with increasing noise ratio. Moreover, NCR (NCR-SCAN) is remarkably superior to SGR (SCAN) in all tests, which shows the generalizability of NCR.

### 4.5.2 Visualization on the Co-divide and Co-rectify

To further investigate the influence of the co-divide and co-rectify modules in our method, we carry out experiments on the Flickr30K dataset by visualizing the per-sample loss distribution and the model predictions on the noisy data, where the noisy ratio is 20%. For better visualization, here we show the result of NCR-SCAN and leave the result of NCR in the supplemental material. As shown in Fig. 3(b), the loss of most noisy samples is larger than the clean loss, which verifies the memorization effect of DNNs. By fitting the per-sample loss with GMM, NCR could effectively divide the data into clean and noisy splits. Regarding the analysis on the co-rectify, Fig. 3(c) shows

---

[3] https://github.com/OpenAI/CLIP
[4] https://github.com/Zasder3/train-CLIP-FT

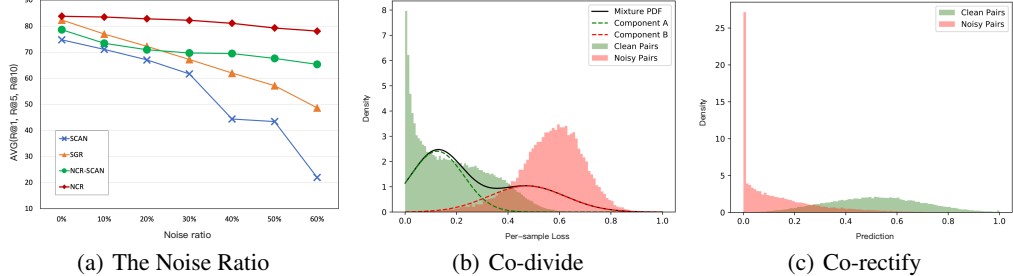

| (a) The Noise Ratio | (b) Co-divide | (c) Co-rectify |

Figure 3: (a) Retrieval performance of NCR and NCR-SCAN on Flickr30K with varying noise ratio. (b) Per-sample loss distribution and GMM fitting visualization after warmup. (c) Model predictions on the noisy subset.

that the rectified soft labels of most clean pairs range into $[0.3, 1]$ and those of most noisy pairs range into $[0, 0.5]$. In other words, one could enforce the similarity of true positives larger than that of the negatives during training, thus eliminating the negative impact of the noisy correspondence.

### 4.5.3 Ablation Study

In this section, we carry out the ablation study on the Flickr30K with the noise ratio of 50%. As shown in Table 4, all these three components are important to achieving encouraging results.

Table 4: Ablation studies on Flickr30K with 50% noise.

| Method | | | Image $\to$ Text | | | Text $\to$ Image | | |
|---|---|---|---|---|---|---|---|---|
| Co-divide | Co-rectify | Warmup | R@1 | R@5 | R@10 | R@1 | R@5 | R@10 |
| ✓ | ✓ | ✓ | 72.9 | 93.0 | 96.3 | 54.3 | 79.8 | 86.5 |
| ✓ | | ✓ | 71.4 | 90.8 | 95.7 | 54.1 | 80.3 | 86.5 |
| | ✓ | ✓ | 16.0 | 38.4 | 51.7 | 12.6 | 31.4 | 42.8 |
| ✓ | ✓ | | 0.3 | 0.6 | 1.0 | 0.2 | 0.5 | 1.1 |

### 4.5.4 Noisy Samples

Fig. 4 shows some noisy CC152K examples identified by NCR. As shown, the first four image-text pairs are completely unrelated, which will be successfully detected by NCR. For the last pair, it will also be detected as noisy correspondence even though the visual and textual modalities are correlated at a coarse-grained level, *e.g.*, both the image and text involve "beach".

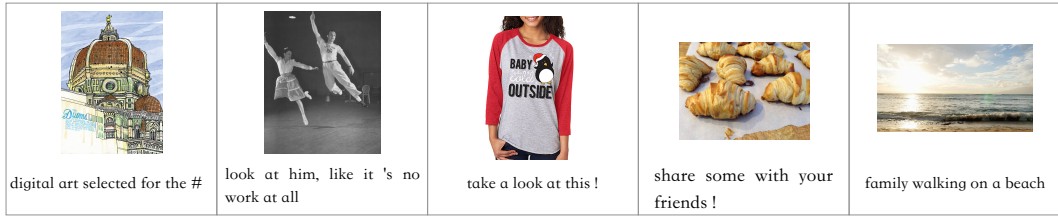

| digital art selected for the # | look at him, like it 's no work at all | take a look at this ! | share some with your friends ! | family walking on a beach |

Figure 4: Some noisy examples correctly divided by NCR.

## 5  Conclusion

This paper could be the first attempt to study a new problem in cross-modal matching, *i.e.*, the noisy correspondence which could be a potential new direction in noise label. To solve this problem in cross-modal matching, we propose rectifying the noisy correspondence by an adaptive prediction function and a novel triplet loss with a soft margin to achieve robust cross-modal matching. Extensive experiments verify the effectiveness of the proposed method in handling synthesized and real noisy correspondences.

## Broader Impact Statement

Cross-modal matching is a fundamental topic in multimodal learning, which could be applied to a wide range of applications including data retrieval, recommender systems, and vision-and-language understanding. This work could be one of the first works to aware of the importance and existence of the noisy correspondence problem in numerous applications. There are many benefits to solving the noisy correspondence problem, *e.g.*, reducing the costs for manually annotating and aligning data; more data could be collected and used even though some of them are incorrectly aligned. Besides the benefits, it should pay attention on the potential negative impacts including but not limited to 1) The risk of automation bias [28] for decision making from the data bias, especially in aviation, health care, and autonomous vehicles. 2) The job loss caused by the NCR since it makes possibility to automatically correct the noisy correspondence, thus remarkably reducing the cost of human labor. We would encourage further work to understand and mitigate the above biases and risks.

## Acknowledgements

The authors would thank to the anonymous reviewers whose valuable suggestions and constructive comments remarkably improve this work. This work was supported in part by the National Key R&D Program of China under Grant 2020AAA0104500; in part by the Key Research and Development Program of Sichuan Province under Grant 2019YFG0497; in part by NFSC under Grant U21B2040, 62176171, U19A2078, 61625204, and 61836006; in part by the Fund of Sichuan University Tomorrow Advancing Life.

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
