# Supplemental Material for Learning with Noisy Correspondence for Cross-modal Matching

**Zhenyu Huang**[*]
College of Computer Science
Sichuan University, China
zyhuang.gm@gmail.com

**Guocheng Niu**
Baidu Inc., China
niuguocheng@baidu.com

**Xiao Liu**
TAL Education Group
liuxiao15@tal.com

**Wenbiao Ding**
TAL Education Group
dingwenbiao@tal.com

**Xinyan Xiao**
Baidu Inc., China
xiaoxinyan@baidu.com

**Hua Wu**
Baidu Inc., China
wu_hua@baidu.com

**Xi Peng**[†]
College of Computer Science
Sichuan University, China
pengx.gm@gmail.com

## 1 Introduction

In this supplemental material, we provide additional information including the network architecture, the training configuration, and implementation details. Then, we conduct additional experiments on MS-COCO (5K valuation) to verify the effectiveness of NCR. To show the effectiveness of the co-divide and co-rectify, we carry out experiments for visualizing the per-sample loss distribution and the model predictions on the noisy data. Furthermore, we visually show some qualitative results of text and image retrieval on CC152K.

## 2 Implementation and Training Details

NCR extends the SGR [2] to be robust again noisy correspondence. Following SGR, for each input image, we extract $K$ region-level visual features with the Faster R-CNN [4] and add a fully-connect layer $f(I)$ to transform them into 1024-dimensional vectors as local representations $\{\mathbf{v}_1, \cdots, \mathbf{v}_K\}$. Then we compute the global representation with self-attention mechanism [6] with $\frac{1}{K}\sum_{i=1}^{K}\mathbf{v}_i$ as query and aggregate all the regions to obtain global representation $\hat{\mathbf{v}}$. For text representation, we split the sentence into $L$ words which are encoded into 300-d vectors by word embedding, then we transform each word into 1024-dimensional vectors with a bi-directional GRU [5] $g(T)$ as local representation $\{\mathbf{t}_1, \cdots, \mathbf{t}_L\}$ and global representations $\hat{\mathbf{t}}$ similar to the image representation.

For the similarity graph reasoning, we compute a similarity vector between given vectors $\mathbf{x}$ and $\mathbf{y}$ as:

$$s(\mathbf{x}, \mathbf{y}; \mathbf{W}) = \frac{\mathbf{W}|\mathbf{x} - \mathbf{y}|^2}{\mathbf{W}\|\mathbf{x} - \mathbf{y}\|^2} \tag{1}$$

where $\mathbf{W}$ is a learnable parameter matrix. With the similarity vector function, we compute the similarity vectors between the images and texts with local and global representations respectively. To

---

[*]Some parts of the work was done while Zhenyu Huang was an internship at Baidu Inc.
[†]Corresponding author.

35th Conference on Neural Information Processing Systems (NeurIPS 2021).

be exact, SGR computes the global similarity as

$$\mathbf{s}^g = \mathbf{s}(\hat{\mathbf{v}}, \hat{\mathbf{t}}; \mathbf{W}^g) \tag{2}$$

and local similarity between $\mathbf{v}_j$ and $\mathbf{t}_j$ as:

$$\mathbf{s}_j^l = \mathbf{s}(\mathbf{a}_j^v, \mathbf{t}_j; \mathbf{W}^l)$$
$$\mathbf{a}_j^v = \sum_{i=1}^{K} \alpha_{ij} \mathbf{v}_i \tag{3}$$

where $\mathbf{a}_j^v$ is the attended visual feature, $\alpha_{ij}$ is the textual-to-visual attention weight similar to [3]:

$$\alpha_{ij} = \frac{exp(\lambda \hat{c}_{ij})}{\sum_{j=1}^{K} exp(\lambda \hat{c}_{ij})} \tag{4}$$

where $\hat{c}_{ij}$ is the normalized cosine similarity between image region feature $\mathbf{v}_i$ and word feature $\mathbf{t}_j$.

Once the similarity vectors $\mathcal{N} = \{\mathbf{s}_1^l, \mathbf{s}_2^l, \cdots, \mathbf{s}_K^l\}$ are obtained, a similarity graph is constructed with edges defined as:

$$e(\mathbf{s}_p, \mathbf{s}_q; \mathbf{W}_{in}, \mathbf{W}_{out}) = \frac{exp((\mathbf{W}_{in}\mathbf{s}_p)(\mathbf{W}_{out}\mathbf{s}_q))}{\sum_q exp((\mathbf{W}_{in}\mathbf{s}_p)(\mathbf{W}_{out}\mathbf{s}_q))} \tag{5}$$

where $\mathbf{W}_{in}$ and $\mathbf{W}_{out}$ are the linear transformations for incoming and outgoing nodes. Finally, we adopt a graph reasoning for final similarity computation. Specifically, we perform similarity graph reasoning by updating the nodes and edges with

$$\hat{\mathbf{s}}_p^n = \sum_q e(\mathbf{s}_p^n, \mathbf{s}_q^n; \mathbf{W}_{in}^n, \mathbf{W}_{out}^n) \cdot \mathbf{s}_q^n$$
$$\mathbf{s}_q^{n+1} = ReLU(\mathbf{W}_r^n \hat{\mathbf{s}}_p^n) \tag{6}$$

where $\mathbf{W}_{in}^n$, $\mathbf{W}_{out}^n$ and $\mathbf{W}_r^n$ are learnable parameters in each step, $\mathbf{s}_p^0$ and $\mathbf{s}_q^0$ are taken from $\mathcal{N}$ at step $n = 0$. SGR iteratively reasons the similarity for $N$ steps, and take the output of the global node at the last step as the reasoned similarity representation, and then feed it into a fully-connected layer to infer the final similarity score, *i.e., $S(I, T)$* in NCR.

Table 1: The parameter settings of training on four datasets.

| Dataset | Warmup Epochs | Epochs | lr_update | batch |
|---|---|---|---|---|
| Flickr30K | 5 | 40 | 30 | 128 |
| MS-COCO | 10 | 20 | 10 | 128 |
| CC152K | 10 | 40 | 30 | 128 |

We also give the training parameters of NCR in three datasets in Table. 1, *i.e.,* warmup epochs, training epochs, learning rate update interval (lr_update) and batch size.

## 3   MS-COCO 5K Validation

To further verify the effectiveness of the proposed method, we conduct the experiments on the full 5K testing of MS-COCO. The results are shown in Table. 2. One could observe that NCR achieves state-of-the-art performance in the noise-free case. When the data is contaminated by the noisy correspondence, NCR remarkably outperforms all the baselines. Specifically, NCR improves R@1 by 3.5%, 3%, 2.2%, and 2.5% in text and image retrieval.

## 4   Evaluation on Many-to-many Correspondence

To show the potential of NCR for modeling the possible many-to-many correspondence between images and captions, we conduct the experiment on the MS-COCO in terms of the Plausible-Match

Table 2: Image-Text Retrieval on MS-COCO 5K.

| Noise Ratio | Methods | Image → Text | | | Text → Image | | |
|---|---|---|---|---|---|---|---|
| | | R@1 | R@5 | R@10 | R@1 | R@5 | R@10 |
| 0% | SCAN | 44.7 | 75.9 | 86.6 | 33.3 | 63.5 | 75.4 |
| | VSRN | 53.0 | 81.1 | 89.4 | 40.5 | 70.6 | 81.1 |
| | IMRAM | 53.7 | 83.2 | 91.0 | 39.7 | 69.1 | 79.8 |
| | SAF | 53.3 | - | 90.1 | 39.8 | - | 80.2 |
| | SGR | 56.9 | - | 90.5 | 40.2 | - | 79.8 |
| | SGRAF | 57.8 | - | **91.6** | **41.9** | - | 81.3 |
| | **NCR** | **58.2** | **84.2** | 91.5 | 41.7 | **71.0** | **81.3** |
| 20% | SCAN | 42.4 | 72.1 | 82.6 | 22.8 | 52.3 | 66.3 |
| | VSRN | 8.9 | 26.5 | 40.2 | 5.7 | 20.3 | 31.4 |
| | IMRAM | 44.3 | 75.5 | 85.7 | 34.1 | 63.1 | 74.5 |
| | SAF | 42.7 | 73.8 | 83.7 | 31.6 | 60.8 | 72.9 |
| | SGR* | 44.6 | 73.5 | 83.7 | 31.4 | 60.4 | 72.4 |
| | SGR-C | 53.4 | 81.5 | 89.3 | 38.4 | 67.8 | 78.8 |
| | **NCR** | **56.9** | **83.6** | **91.0** | **40.6** | **69.8** | **80.1** |
| 50% | SCAN | 18.5 | 44.5 | 58.9 | 2.2 | 6.2 | 9.6 |
| | VSRN | 8.3 | 25.4 | 37.7 | 4.8 | 18.1 | 29.2 |
| | IMRAM | 5.0 | 23.0 | 38.5 | 8.1 | 26.0 | 38.3 |
| | SAF | 10.4 | 32.8 | 48.2 | 15.2 | 38.3 | 51.9 |
| | SGR* | 36.4 | 64.8 | 77.1 | 26.0 | 52.9 | 64.3 |
| | SGR-C | 50.1 | 77.4 | 86.8 | 35.4 | 64.5 | 76.0 |
| | **NCR** | **53.1** | **80.7** | **88.5** | **37.9** | **66.6** | **77.8** |

R-Precision (PMRP) proposed by [1]. PMRP provides a comprehensive evaluation of the image-text retrieval in the scenario of possible many-to-many correspondence between images and captions. The results are shown in Table. 3. One could observe that the proposed method NCR still outperforms the SGR with a large margin on the PMRP metric, showing the powerful capacity of NCR for revealing the many-to-many correspondence.

Table 3: Comparison on MS-COCO 1K in terms of PMRP.

| Noise Ratio | Methods | Image → Text | | | | Text → Image | | | |
|---|---|---|---|---|---|---|---|---|---|
| | | R@1 | R@5 | R@10 | PMRP | R@1 | R@5 | R@10 | PMRP |
| 20% | SGR | 67.8 | 91.7 | 96.2 | 22.0 | 52.9 | 83.5 | 90.1 | 23.9 |
| | **NCR** | **77.7** | **95.5** | **98.2** | **47.2** | **62.5** | **89.3** | **95.3** | **47.7** |
| 50% | SGR | 60.6 | 87.4 | 93.6 | 16.5 | 46.0 | 74.2 | 79.0 | 18.8 |
| | **NCR** | **74.6** | **94.6** | **97.8** | **46.2** | **59.1** | **87.8** | **94.5** | **46.7** |

## 5 Visualization Experiments

In this section, we first investigate the effect of co-divide and co-rectify of NCR with the visualization results. Then, we visually demonstrate some retrieval results on CC152K.

### 5.1 Visualization on the Co-divide and Co-rectify

In the main paper, we carry out experiments on the Flickr30K dataset by visualizing the per-sample loss distribution and the model predictions on the noisy data. We only show the results of NCR-SCAN for better visualization, here we provide the results from NCR in Fig. 1(a) and Fig. 1(b). From Fig. 1(a), one could observe that the loss of most noisy samples is larger than the clean loss, which verifies the memorization effect of DNNs. Regarding the analysis on the co-rectify, Fig. 1(b) shows

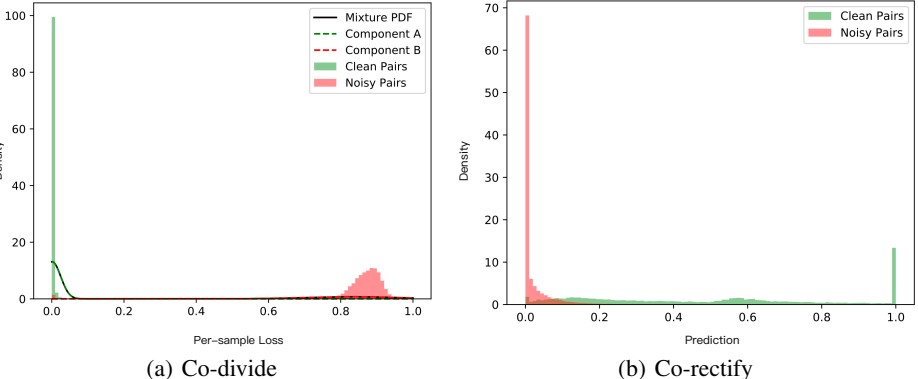

(a) Co-divide        (b) Co-rectify

Figure 1: (a) Per-sample loss distribution and GMM fitting visualization after warmup. (b) Model predictions on the noisy subset.

that the rectified soft labels of most clean pairs range into $[0.2, 1]$ and those of most noisy pairs range into $[0, 0.2]$.

## 5.2 Retrieval Results

We show qualitative results of NCR for text-to-image retrieval in Fig. 2, and image-to-text retrieval in Fig. 3. One could observe that NCR could correctly retrieval the corresponding image/text with given text/image, as shown Fig. 2 (1)–(2) and Fig. 2 (1)–(3). In addition, we also give some failure cases, as shown Fig. 2 (3)–(4) and Fig. 3 (4). It is interesting and surprising that in Fig. 2 (4), the retrieval image by NCR is more in line with the text description (*i.e.,* the stream flows through the rocks) compared to the ground truth. Such a case shows that some image-text pairs are matched inaccurately and some images/text may have more than one matched text/images in the dataset.

(1) koi swimming in a swirl             (2) little girl riding a toy car

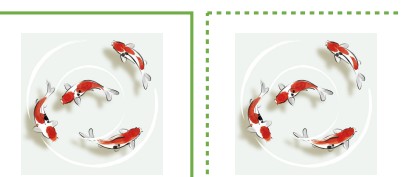        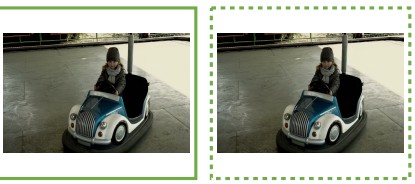

(3) the country highlighted in red on an abstract illustrated map of the continent      (4) the stream flows through the rocks

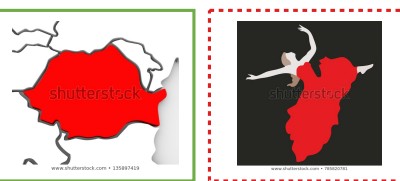       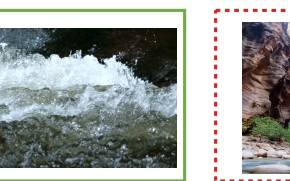 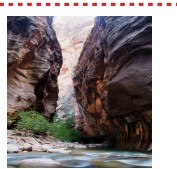

Figure 2: Some examples in text-to-image retrieval. In each example, the left picture is the ground truth while the right picture is the retrieval image by NCR. We outline the true matched images in green dashed boxes and the false matched in red dashed boxes.

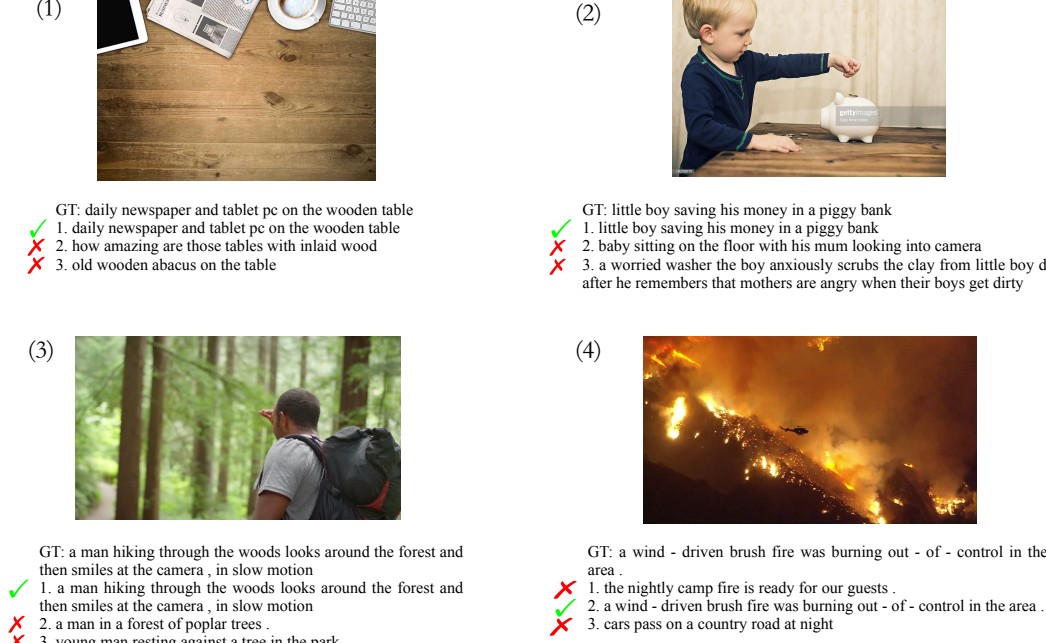

Figure 3: Some examples in image-to-text retrieval. In each example, the first sentence is the ground truth text and the rest sentences are the top 3 ranked retrieval text. We mark the true matched text with ✓ and the false matched images with ✗.