# OpenReview forum: "Learning with Noisy Correspondence for Cross-modal Matching"
_NeurIPS.cc/2021/Conference — NeurIPS 2021 Oral_

### Official Review · Reviewer_nty3 · 2021-07-12

**Rating:** 9
**Confidence:** 5

**Summary:**

This paper proposes a noisy label learning framework for cross-modal learning, where the training data consists of wrongly matched modalities. The proposed NCR first divides the data into clean and noisy splits based on the difference of loss distribution. Then NCR proposes to rectify the labels with a prediction function and further recast them into soft margins in the triplet loss. Extensive experiments and comparisons are conducted to verify the effectiveness of the method.

**Limitations And Societal Impact:**

The paper provides a comprehensive discussion about the potential negative impact in the Border Impact Statement.

**Main Review:**

Overall, the paper is generally well written and easy to read. The main strengths and weaknesses of this paper are summarized as follows.

Strengths：
1.	The experiments are comprehensive. To show the effectiveness of the proposed method, the authors take SGR as an example and provide comparisons with other SOTA methods. The results with varying noise ratios and real-world noisy data demonstrated the effectiveness of NCR. Besides the SGR, experimental results of NCR-SCAN also show the superiority of the proposed method.
2.	the technique presented in the paper is novel. To achieve the robust cross-modal matching, the authors propose to turn the rectified soft labels into soft margins which enforce the true positive pairs closer than the negative by a large margin, while the false positive will has a small margin.

Weaknesses:
1.	It seems that the proposed method NCR is only applicable to the triplet loss in cross-modal matching as defined in Eq.5-6. How to achieve robustness to other loss formulations like the softmax loss in ALIGN (Scaling Up Visual and Vision-Language Representation Learning With Noisy Text Supervision)?
2.	NCR recast the soft labels as soft margins defined in Eq.7. Why NCR need this procedure for robust matching? Why it could achieve robustness?
3.	Since two models are trained individually in the manner of co-teaching. It inevitably needs more computation time for convergence. And how to obtain the final retrieval results with two models?


**Time Spent Reviewing:**

3

---

> ### Author Response · Authors · 2021-08-10
> **The Response to Reviewer nty3**
>
> Thanks for your comments. We will answer the questions one by one.
>
> **Q1**: It seems that the proposed method NCR is only applicable to the triplet loss in cross-modal matching as defined in Eq.5-6. How to achieve robustness to other loss formulations like the softmax loss in ALIGN.
>
> NCR could be extended to other forms of matching loss. In the submission, we take the current SOTA method SGR (SGRAF) as a showcase to achieve the robust cross-modal matching, in which NCR predicts the pairs in Eq.5 and recasts the soft labels into triplet loss in Eq.6. It is easy to extend NCR to another matching/metric learning loss. Specifically, for the prediction function, it could be designed as the normalized similarity scores in other matching frameworks similar to NCR. For the soft labels, some alternatives propose a static margin to boost performance such as CosFace [ref.A] and ArcFace loss [ref.B]. Similar to triplet loss used in NCR, we could recast the soft labels into the soft margins in softmax losses to achieve robust cross-modal matching. A more intuitive way is simply assigning a weight to each sample as the soft labels to eliminate the negative impact from noisy pairs.
>
> **Q2**: NCR recast the soft labels as soft margins defined in Eq.7. Why NCR need this procedure for robust matching? Why it could achieve robustness?
>
> To utilize the soft labels rectified by the co-rectify module, NCR needs to recast soft labels as soft margins into triplet loss as the conventional triple only adopt binary pairs (positive or negative pairs). By incorporating the soft labels into the matching loss, we assign large margins to the clean pairs while small margins to the noisy ones such that the uncertainty can naturally be captured in the embedding space to achieve robustness (as also pointed out by reviewer 1).
>
> **Q3**: Since two models are trained individually in the manner of co-teaching. It inevitably needs more computation time for convergence. And how to obtain the final retrieval results with two models?
>
> Yes. To train the two models for co-divide and co-rectify, NCR needs additional time (usually double training time). The inference implementation detail could refer to section 4.1. In brief, we perform the image-text retrieval based on the average similarity computed by two models.
>
>
>
> **References:**
>
> [A] Wang, H., Wang, Y., Zhou, Z., Ji, X., Gong, D., Zhou, J., ... & Liu, W. (2018). Cosface: Large margin cosine loss for deep face recognition. In *Proceedings of the IEEE conference on computer vision and pattern recognition* (pp. 5265-5274).
>
> [B] Deng, J., Guo, J., Xue, N., & Zafeiriou, S. (2019). Arcface: Additive angular margin loss for deep face recognition. In *Proceedings of the IEEE/CVF Conference on Computer Vision and Pattern Recognition* (pp. 4690-4699).

---

> > ### Comment · Reviewer_peXt · 2021-08-16
> > **Response to authors' rebuttal**
> >
> > The response has addressed my concerns well by explaining the practical meaning of the proposed soft margins and the implicit assumption on noisy data (at least 10% clean) for label predictions.
> >
> > Overall, I agree that learning with noisy correspondence is a new and promising direction by considering the label errors in the pair-form data, i.e. the noisy correspondence or mismatching. This new problem has never been studied before as far as I known. I think the work could provide novel insight to the community and the revealed noisy correspondence is important and common in a wide range of one-to-one applications.
> >
> > Regarding the technical controbution of this paper, the author takes image-text retrieval as a showcase and verifies the effectiveness of the proposed NCR method comparing to other SOTA methods. The experiments on the challenging dataset (e.g., CC152K) have shown the superiority of NCR in handling real-world noisy data. In addition, I also noticed that the authors have provided additional results with CLIP which is a SOTA method in many tasks in another response. The experimental results show the superiority of NCR even compared to the large model (CLIP) pretrained in massive data.
> >
> > In summary, I think the result of the work is solid and the idea (i.e., noisy correspondence) is a new direction to the community.

---

> > > ### Author Response · Authors · 2021-08-24
> > > **Response to the Final Comment**
> > >
> > > We sincerely thank you for your positive recognition and evaluation of our work and will correct the typos in the next version.

---

> ### Comment · Reviewer_nty3 · 2021-08-16
> **Rating increased due to satisfactory author response**
>
> I would like to thank the authors for their response to my review. I found that the authors provided enough information to my initial questions regarding to 1) robustness to other loss formulations; 2) why soft margin is used (I like the idea of assigning large margins to the clean pairs and small margins to the noisy ones); 3) the computation time and more detail about the co-teaching. Therefore, I increased my rating.

---

> > ### Author Response · Authors · 2021-08-24
> > **Response to the Final Comment**
> >
> > Thanks for your positive comments. We would consider adding more discussion to clarify the roles of the proposed losses and soft margins in the next submission.

---

### Official Review · Reviewer_peXt · 2021-07-13

**Rating:** 9
**Confidence:** 5

**Summary:**

This paper focuses on cross-modal matching with the presence of noisy correspondence. The author proposes a novel method for learning with noisy correspondence to achieve robust cross-modal matching based on the SGR. In brief, it first splits the data into clean and noisy subsets in the manner of co-teaching and rectifies the labels with an adaptive prediction function. Extensive experiments on three datasets show the effectiveness of the proposed method.

**Limitations And Societal Impact:**

The authors have discussed the limitation and potential negative impact of the proposed method.

**Main Review:**

In brief, this paper did a good job for solving the noisy problem in cross-modal matching. The proposed method is well verified with dense experiments on three benchmark datasets. Here are some detailed comments.
Pros：
1.	The proposed method for learning with noisy correspondence is novel. To perform label refinement in the scenario of a matching problem, the author proposes an adaptive prediction function for label rectifying. Moreover, to incorporate the rectified soft labels in the triplet loss, they propose to recast the labels as the soft margin in the loss.
2.	Experiments on the simulation noisy datasets (COCO and Flickr) and the real-world dataset (Conceptual Captions) show the superiority of the proposed method. Moreover, the ablation study and visualization verify the effectiveness of different modules.
Cons:
1.	In the method, the soft labels are recast as soft margins in the triplet loss. Why are the soft margins computed in the form of Eq. 7? Are there other alternative ways to determine the soft margins?
2.	There is a typo in Eq. 5. S is the similarity margin between the given pair and the mean of negatives. The second minus symbol should be modified as a plus.
3.	The model predictions are based on the average similarity margin of the largest 10% pairs in Eq.6. How to guarantee that the predictions are accurate enough for label rectifying? Why only select 10% pairs?
4.	I’m a bit concerned about training time as the NCR trains two models in the manner of co-teaching.


**Time Spent Reviewing:**

3

---

> ### Author Response · Authors · 2021-08-10
> **The Response to Reviewer peXt**
>
> Thanks for your comments. We will answer the questions one by one.
>
> **Q1**: In the method, the soft labels are recast as soft margins in the triplet loss. Why are the soft margins computed in the form of Eq. 7? Are there other alternative ways to determine the soft margins?
>
> Eq. 7 is designed to assign a small margin to the pairs when $\hat{y}_i$ is close to 0, while a large margin to the pairs when $\hat{y}_i$ is close to 1. Moreover, as shown in the curve visualization in Fig.2(b), the soft margins are increasing **exponentially** when  $\hat{y}_i$ approaches to 1, further enforcing the "clean'' positive pairs much closer than the "noisy" ones. The alternative ways of defining the soft margins are feasible as they could assign larger margins to the clean positive pairs and small margins to the noisy ones.
>
> **Q2**: There is a typo in Eq. 5. S is the similarity margin between the given pair and the mean of negatives. The second minus symbol should be modified as a plus.
>
> Thanks. We will correct it in the revision paper.
>
> **Q3**: The model predictions are based on the average similarity margin of the largest 10% pairs in Eq.6. How to guarantee that the predictions are accurate enough for label rectifying? Why only select 10% pairs?
>
> The predictions are computed based on the average similarity margin of the largest 10% pairs which could be regarded as the measurement anchor to guarantee the precision of predictions. We select the 10% most similar pairs as the clean ones to determine whether the other pairs are positive or negatives. This is based on a simple assumption that the training data (image-text pairs) has at least 10% clean samples which are easily satisfied in practice. The experimental results on three benchmark datasets with varying noise ratios also verify the effectiveness of this assumption.
>
> **Q4**: I’m a bit concerned about training time as the NCR trains two models in the manner of co-teaching.
>
> Yes. Since NCR models the noisy data in the manner of co-teaching, it inevitably introduces more computation cost for additional network training.

---

### Official Review · Reviewer_iPic · 2021-07-16

**Rating:** 8
**Confidence:** 5

**Summary:**

This paper introduces a novel task, noisy correspondence of cross-modal retrieval tasks. The noisy correspondence is similar to the noisy label in object classification tasks, where the noisy correspondence scenario assumes that some image-caption pairs are incorrectly matched. To solve the problem, this paper proposes a Noisy Correspondence Rectifier (NCR), which could be considered as the novel extension of Co-teaching [9] to cross-modal retrieval tasks. Note that directly applying Co-teaching to cross-modal retrieval is non-trivial as the authors discussed in the paper (L105-115), because noisy label methods assume the existence of the prediction score function of the given input, while cross-modal retrieval methods usually do not have such per-sample score function. Hence, the proposed NCR first selects (co-divides) clean and noisy samples by using triplet loss values and the 2-mean Gaussian Mixture Model (GMM). Using the divided samples, NCR co-rectifies the noisy labels by the prediction function, where the prediction function is defined by the value of triplet distances. Using the extracted clean/noisy samples and rectified labels, NCR applies the hardest negative mining triplet loss to train the model, while the margin value is adaptive to the rectified labels. The experimental results show the effectiveness of NCR in terms of robustness to the noisy correspondence, e.g., achieving nearly dropped retrieval performances when 50% of correspondences are incorrect.

**Limitations And Societal Impact:**

One possible limitation of this paper is the lack of discussions on the existing noisy correspondence in the existing cross-modal retrieval methods. I would suggest adding more related works and their discussions in the main paper.

Another possible limitation could be the evaluation metric; Recall@K is too noisy and less informative metric than precision. I suggested the authors testing PMRP proposed by [A] in MS-COCO benchmark for measuring precision performances by the proposed method.

**Main Review:**

## Pros

- (+) The introduced noisy correspondence is an interesting and important problem. Although a few previous works [A, B] tackle similar motivation in cross-modal retrieval tasks, as far as the reviewer knows, this paper is the first paper that evaluates synthetic noisy correspondence with state-of-the-art cross-modal retrieval methods.
    - [A] Chun, Sanghyuk, et al. "Probabilistic embeddings for cross-modal retrieval." Proceedings of the IEEE/CVF Conference on Computer Vision and Pattern Recognition. 2021.
    - [B] Wray, Michael, Hazel Doughty, and Dima Damen. "On Semantic Similarity in Video Retrieval." Proceedings of the IEEE/CVF Conference on Computer Vision and Pattern Recognition. 2021.
- (+) The proposed approach is sound and empirically achieves good results under the noisy correspondence scenario. Especially, The proposed Co-teaching-like algorithm is conceptually well-founded and empirically works well.
- (+) Experimental results on different noise levels (0%, 20%, 50%) are very interesting; particularly, existing methods perform poorly when noisy correspondence happens.
- (+) This paper is well-written, well-organized, and easy to follow. I really enjoyed reading this paper.

## Cons

I think this paper has a good problem definition contribution and technical contribution. However, I feel that discussions related to the previous attempts (could be considered as contemporary works) are not enough. Especially if the authors can add discussions related to Chun et al. [A], I think this paper will be stronger submission.

### Precision metric

Musgrave et al. [C] recently showed that "the use of accuracy metrics that are either misleading, or do not a provide a complete picture of the embedding space", in other words, recall scores may lead to a wrong conclusion. Chun et al. [A] tackles the same problem in cross-modal retrieval tasks, and proposed Plausible-Match R-Precision (PMRP) for MS-COCO evaluation. I suggest the authors report PMRP scores [A] to show the effectiveness of the proposed NCR method. As far as the reviewer understood, re-evaluating models with PMRP scores does not need extra training budgets but only needs the evaluation procedure which is relatively reasonable computation during the rebuttal period.

- [C] Musgrave, Kevin, Serge Belongie, and Ser-Nam Lim. "A metric learning reality check." European Conference on Computer Vision. Springer, Cham, 2020.

### Possible related works

As far as the reviewer understood, this paper mainly assumes the underlying dataset has (almost) clean correspondence. However, in practice, the existing widely-used cross-modal datasets, such as MS-COCO already have many noisy correspondences. For example, many images in MS-COCO only contain a big wave and a surfer and their corresponding captions are barely distinguishable (See Figure 4 of [A] as another example). A similar problem, many-to-many correspondence, is actively discussed in Chun et al. [A], which solves the problem by a probabilistic approach. I believe that [A] should be discussed in the main paper because the target problem by [A] is highly related to the proposed noisy correspondence problem. This could be an explanation of why NCR performs better than other methods in 0% noise scenarios in Table 1 and 2. Wray et al. [B] also targeted a similar problem, instance-based retrieval evaluation of video retrieval, and solved the problem by soft labels as rectified soft labels in this paper. Kim et al. [D] also employed soft non-binary labels to solve metric learning tasks.

- [D] Kim, Sungyeon, et al. "Deep metric learning beyond binary supervision." Proceedings of the IEEE/CVF Conference on Computer Vision and Pattern Recognition. 2019.

The listed related works do not hurt the novelty of this paper, but I think adding discussions on these related works, the submission will be stronger.

### Impact of vanilla triplet (or warmup)

- In L191-197, the paper mentions that $L_\text{soft}$ uses the hardest negative mining, not the vanilla triplet loss without hard negative mining as "L_w". I wonder about the performance gap between the vanilla triplet and the hardest negative mining triplet while L192-193 says "Although the hard negatives are helpful in improving the performance".
- On the other hand, it seems that the vanilla triplet loss without hard negatives hurt the performance of SGR in CC152k, as shown in Table 2. Why does it happen? The paper assumes that SGR is very sensitive to noisy correspondence (as denoted in L246-249), but there is no explanation of how the vanilla triplet can mitigate the noisy correspondence.
  - Similarly, warm-up (with the vanilla triplet) seems the most important ingredient of the method (Table 3). It may imply that the vanilla triplet is one of the most important keys to solve the noisy correspondence. I wonder about the performance of warmup only, i.e., the vanilla triplet trained model.

### More related works

While the following related works do not hurt the novelty of this paper, I suggest the authors add more discussions on previous works [A, B] discussing noisy correspondence problems as well. Especially,


## Comments not affecting the decision score

- L143-L149 can be replaced by citation of the textbook by Bishop, Christopher M (Bishop, Christopher M. "Pattern recognition." Machine learning 128.9 (2006).)
- L164: "On the other hand" appears two times in sequential sentences. It could be a typo.
- L186: It would be good to add VSE++ citation for explaining the hardest negative mining strategy
- Table 3. "02." could be a typo of "0.2"

## Overall comment

This paper introduces an important and undiscovered problem, noisy correspondence. The proposed method motivated by noisy label methods successfully tackles the problem. Although there are some raised concerns, for example, (1) missing precision comparisons in MS-COCO (2) some missing related works (3) study on vanilla triplet without hard negative mining is missing, I think these issues are not a significant issue to reject this paper, but resolving the raised concerns will make the submission stronger. Overall, I vote for accepting this paper.

---

## Post-rebuttal update

I really appreciate the authors addressing my suggestions very well in the response. The precision results (outperforming the baseline with more than +20%) are particularly interesting. I revise my score to clear accept.

**Time Spent Reviewing:**

6

---

> ### Author Response · Authors · 2021-08-10
> **The Response to Reviewer iPic**
>
> Thanks for your review and suggestions! We will answer the questions one by one.
>
> **Q1**: Missing precision metric (PMRP).
>
> As suggested, we have added the experiments on the MS-COCO dataset in terms of Plausible-Match R-Precision (PMRP) to further show the effectiveness of NCR. PRMP is proposed by [ref.A] to provide a comprehensive evaluation of the image-text retrieval in the scenario of possible many-to-many correspondence between images and captions. We use the source code at https://github.com/naver-ai/pcme to compute the PRMP score. The experimental results are as follows:
>
> | MS-COCO 1K Test (20% Noise) | I2T  | I2T  | I2T  | I2T  | T2I  | T2I  | T2I  | T2I  |
> | --------------------------- | :--: | :--: | :--: | :--: | :--: | :--: | :--: | :--: |
> |                             | R@1  | R@5  | R@10 | PMRP | R@1  | R@5  | R@10 | PMRP |
> | SGR                         | 67.8 | 91.7 | 96.2 | 22.0 | 52.9 | 83.5 | 90.1 | 23.9 |
> | NCR                         | 77.7 | 95.5 | 98.2 | 47.2 | 62.5 | 89.3 | 95.3 | 47.7 |
>
> | MS-COCO 1K Test (50% Noise) | I2T  | I2T  | I2T  | I2T  | T2I  | T2I  | T2I  | T2I  |
> | --------------------------- | :--: | :--: | :--: | :--: | :--: | :--: | :--: | :--: |
> |                             | R@1  | R@5  | R@10 | PMRP | R@1  | R@5  | R@10 | PMRP |
> | SGR                         | 60.6 | 87.4 | 93.6 | 16.5 | 46.0 | 74.2 | 79.0 | 18.8 |
> | NCR                         | 74.6 | 94.6 | 97.8 | 46.2 | 59.1 | 87.8 | 94.5 | 46.7 |
>
> Due to the time limitation, we only report the results of SGR (one of the best baseline) and NCR, and the PMRP scores are computed only on the 1K MS-COCO test data. From the table, we could observe that our method NCR still outperforms SGR in the new metric PMRP, indicating the effectiveness of NCR in revealing underlying many-to-many correspondence in image-text pairs.
>
>
>
> **Q2**: Missing related works.
>
> Thanks for pointing out the missing related works. The next version will add the discussions with these works, and a brief version is as below:
>
> We note that [ref.A] introduces a new paradigm for cross-modal matching, i.e. possible many-to-many correspondence that existed in the image and captions. To achieve the end, the probabilistic representations are learned to model the one-to-many correspondence. Different from [ref.A] which focuses on revealing the possible many-to-many correspondence between image and captions, NCR introduces the noisy correspondence problem and proposes to eliminate the negative impact from noisy pairs. In addition, NCR may reveal some possible non-labeled correspondence mentioned by [ref.A], which explains the success of NCR in non-noise settings (0% noise).
>
> In recent, [ref.B] and [ref.C] propose to employ soft labels in cross-modal retrieval and metric learning methods. More specifically, [ref.B] recasts the soft similarity into binary labels by directly setting a threshold on the predicted similarity. [ref.C] proposes a log-ratio matching loss with a regularization defined by the label distance ratio, which is computed by the continuous labels. Different from them, NCR proposes to recast the rectified soft labels into soft margins in the triplet loss by assigning large margins to the true positive pairs and small ones to the false positive pairs. Moreover, another major difference between our loss and these two works is that the former is specifically designed to solve the noisy correspondence problem whereas the latter did not.
>
>
>
> **Q3**: Impact of vanilla triplet (or warmup)
>
> (i) The performance gap between the vanilla triplet and the hardest negative mining triplet in $L_{soft}$ is as below:
>
> | Flickr30K 50% noise                        | I2T  | I2T  | I2T  | T2I  | T2I  | T2I  |
> | ------------------------------------------ | ---- | ---- | ---- | ---- | ---- | ---- |
> |                                            | R@1  | R@5  | R@10 | R@1  | R@5  | R@10 |
> | NCR (without hard negatives in $L_{soft}$) | 58.5 | 83.0 | 89.5 | 41.8 | 66.6 | 74.6 |
> | NCR (with hard negatives in $L_{soft}$​)    | 72.9 | 93.0 | 96.3 | 54.3 | 79.8 | 86.5 |
>
> (ii) We assume that the reviewer might misunderstand the results in CC152K. In fact, SGR* (trained without hard negatives in the first 10 epochs, and then with hard negatives in the last 10 epochs) is superior instead of inferior to the original SGR (only trained with hard negatives).
>
> The reason why the vanilla triplet can mitigate the noisy correspondence is that the vanilla triplet loss enforces the similarity of given pairs larger than the **average similarity** of all negative pairs in a mini-batch with a margin. Even the data is noisy correspondence, the vanilla triplet would not degrade the matching performance since the false positives are enforced to closer comparing with the average negatives.
>
> In Table 3, the warmup step seems crucial to NCR, which could attribute to the that the warmup provides an initial network for co-divide and co-rectify. Once the initial network is contaminated, the following co-divide and co-rectify steps will magnify and accumulate the errors, thus leading to undesirable matching results. As suggested, we present the experiments of NCR with the warmup step only as below:
>
> | MS-COCO 1K Test (20% Noise) | I2T  | I2T  | I2T  | T2I  | T2I  | T2I  |
> | --------------------------- | ---- | ---- | ---- | ---- | ---- | ---- |
> |                             | R@1  | R@5  | R@10 | R@1  | R@5  | R@10 |
> | NCR (Warmup Only)           | 54.4 | 86.9 | 94.2 | 47.1 | 81.4 | 90.8 |
> | NCR                         | 77.7 | 95.5 | 98.2 | 62.5 | 89.3 | 95.3 |
>
> | MS-COCO 1K Test (20% Noise) | I2T  | I2T  | I2T  | T2I  | T2I  | T2I  |
> | --------------------------- | ---- | ---- | ---- | ---- | ---- | ---- |
> |                             | R@1  | R@5  | R@10 | R@1  | R@5  | R@10 |
> | NCR (Warmup Only)           | 45.1 | 80.0 | 90.9 | 39.2 | 75.3 | 86.5 |
> | NCR                         | 74.6 | 94.6 | 97.8 | 59.1 | 87.8 | 94.5 |
>
>
>
> **Q4**: Typos.
>
> Will correct it with many thanks.
>
>
>
> **References:**
>
> - [A] Chun, Sanghyuk, et al. "Probabilistic embeddings for cross-modal retrieval." Proceedings of the IEEE/CVF Conference on Computer Vision and Pattern Recognition. 2021.
>
> - [B] Wray, Michael, Hazel Doughty, and Dima Damen. "On Semantic Similarity in Video Retrieval." Proceedings of the IEEE/CVF Conference on Computer Vision and Pattern Recognition. 2021.
>
> - [C] Kim, Sungyeon, et al. "Deep metric learning beyond binary supervision." Proceedings of the IEEE/CVF Conference on Computer Vision and Pattern Recognition. 2019.
>
> - [D] Faghri, F., Fleet, D. J., Kiros, J. R., & Fidler, S. (2017). Vse++: Improving visual-semantic embeddings with hard negatives. *arXiv preprint arXiv:1707.05612*.

---

> > ### Comment · Reviewer_iPic · 2021-08-23
> > **Rebuttal response**
> >
> > Dear authors, I just revised my score to "clear accept". I really thank the authors for addressing my suggestions very well in the rebuttal. Please see my "Post-rebuttal update" for the detailed comment.

---

> > > ### Author Response · Authors · 2021-08-24
> > > **Response to the Final Comment**
> > >
> > > Thanks for your positive comments and suggestions. We would include the results of MS-COCO in terms of PMRP for evaluation in revealing possible many-to-many correspondence. Moreover, the discussion with the referred works, i.e., many-to-many correspondence and soft triplet loss, will be added in the next version as suggested.

---

### Official Review · Reviewer_9fra · 2021-07-16

**Rating:** 7
**Confidence:** 5

**Summary:**

The authors propose a method for cross-modal retrieval in the presence of noisy image-text pairings. While there has been a lot of work in learning in the presence of noise in this space, existing works mainly handle the problem of "label" noise - i.e. when semantic categories / labels on the data are available, but are noisy. In contrast, the authors tackle the unsupervised setting where all that is available are image-text pairs harvested from the web. Some of these pairs may be noisy and some may not be. The idea of the paper is straightforward. The authors use a technique commonly used to address noisy labels - i.e. co-teaching. They train two networks, A and B on the same data. After each epoch, the authors collect the similarity scores computed by each of the networks and use it to train a Gaussian Mixture Model. The reason for this is the GMM allows one to calculate the probability that a sample is from a given distribution. They model it with two Gaussians. The idea is that the GMM will fit one to the set of data that has lower similarity (hence noisy) and a separate Gaussian to the one with high similarity (hence clean). They then use the learned similarities predicted by each model to "correct" the labels used for training the other model (for example, treating some data that was considered clean as noisy or vice versa) based on the co-teacher's predictions. Finally, they integrate the similarity / confidence that the model is clean or not into the triplet loss as an adaptive margin, rather than use a static margin. The authors experimentally validate their method on MSCOCO, FlickR30K, and Conceptual Captions and compare against a number of baselines showing reasonable improvement.

**Ethical Concerns:**

This is a technical retrieval work and has no dataset contribution. There are no ethical concerns beyond any existing cross-modal retrieval model.

**Limitations And Societal Impact:**

This is a technical retrieval work and has no dataset contribution. There are no societal concerns beyond any existing cross-modal retrieval model.

**Main Review:**

# Strengths
The problem tackled by the authors is a good one - many cross-modal retrieval methods that exist assume some sort of label space is either available, or can be predicted on the data. However, this might not always hold, particularly when one is harvesting web-scale data without labels. Thus, the problem of noisy pairings is important to address. I have done a literature search and haven't found any papers explicitly tackling this problem directly, so I would agree that the problem itself does have some novelty (although, many other papers do learning in the presence of label noise, which is somewhat different).

The approach taken by the authors is interesting, as they train two models, then try to essentially learn based on the distribution of the learned similarities which are the noisy samples and which are the clean ones - this is essentially a clustering, but done using a GMM. The authors don't have any novelty in that regard though, as the GMM is a standard off-the-shelf model. However, they use it to essentially get a per sample score of how likely the sample is noisy or not. The main idea is that these are then used to update the training set of the other model being used for co-teaching. This is a good idea, as it extends co-teaching methods into the cross-modal retrieval space in the absence of any labeled data.

The authors' way of integrating this knowledge into the model is an interesting one - they essentially use an adaptive margin in the triplet loss. The main idea is that when we are really confident about a sample being clean, we should enforce a large margin. But when we aren't confident the sample is clean or noisy, we should enforce a small margin. This is a good idea, as it allows the model to learn an embedding space where such uncertainty can naturally be captured.

Experimentally, the authors show reasonable gains in the presence of noise. For the MS-COCO dataset, in the most noisy setting (50% pairings), the authors show ~2.1% gain for T-->I and around ~2.9% gain for I-->T on COCO. However, they are often outperformed in the 0% noise setting. The authors show more convincing gains on the Conceptual Captions dataset.

The authors include an ablation where they remove various components of their method, such as various pieces of the co-teaching model and show that each has a contribution to improving the performance of the method.

# Weaknesses
Before getting into specifics of the details of the model, I would like to point out there is an implicit assumption made by the authors that unclean pairs will end up having a lower similarity after the initial training, while clean pairs will have a higher similarity. The authors note throughout the text that models tend to pick up obvious correspondences first, for example. Thus, the assumption is, if samples have a low similarity, they could possibly be unclean (mismatched image-text). I can buy this argument on some datasets, like COCO and Flickr, where there are clear, defined object categories and objects mentioned in the captions. In this case, the model can quickly pick up the correspondences in caption between "car" and the visual object "car" (for example). Thus it makes sense that after the initial training such samples will have a high similarity, while noisy samples will have a lower similarity. However, I would argue this assumption is not always applicable - and **most importantly**, it is not applicable to the problem setting the authors' method is designed to address. Specifically, for datasets like Coco and Flickr or other datasets where there is a strong grounding between image and caption, for these types of datasets there *are* often object labels or categories available, or, if not, they can be predicted using many of the other existing methods for learning cross-modal retrieval in the presence of noisy labels (you can, for example, identify a set of objects in the captions, train a model to classify them, and then do some prediction of labels - and voila, you have a self-supervised and noisy set of labeled data). However, the situation that the authors' method is most applicable to are cases where a label set of objects or classes cannot be readily defined, because obviously if such a label set can be defined there are many baselines that deal with that. So, the authors approach is best attuned to situations where a label set can't be defined, like when harvesting a ton of data from the web, with very loose image-text pairings and an unknown set of classes. If the authors wish to argue their method is applicable to the cases where labels can be defined, then they should show results comparing against those noisy self-supervised label methods. But assuming they don't, then the most important dataset the authors test on is the Conceptual Captions dataset. This dataset does not have a definable set of labels, as it is just a huge set of images and paired text crawled from the web. Again, that's the setting the authors are targeting. However, this type of data exhibits properties that break the authors' assumptions. For example, real world data often features images and text that aren't directly overlapping (i.e. the text isn't really a caption, but just goes with the image) - in other words, the image and text are complementary with one another. In other words, the text and image are related, but not directly duplicative of meaning. This is an important aspect of such data to capture, but the authors approach will actually de-emphasize it. For example, after training, these samples exhibiting this complementarity will have a lower similarity score than the obvious caption-like samples, thus the authors' approach may discard them as noisy.
There is a recent work that tackles this exact scenario in ECCV20 and doesn't require any labeled data and learns a strong semantic space. I am also interested in hearing the authors thoughts on they can distinguish noisy samples from samples that just have a more indirect relationship between image and text (the so-called complementarity). This is a very challenging problem. Moreover, the authors should have compared against this paper. In my view, the results on FlickR and COCO aren't really the scenario this method is suitable for.
Paper studying complementarity and dealing with indirect relationship of image-text on real world data:
Thomas, Christopher, and Adriana Kovashka. "Preserving semantic neighborhoods for robust cross-modal retrieval." European Conference on Computer Vision. Springer, Cham, 2020.

Again, the authors paper is targeting learning on web-scale data where there are no discernable classes and thus there is noise. Thus, I would expect the authors to compare against using CLIP as a baseline. CLIP is trained on a massive dataset of webly harvested data and presumably there are a lot of image-text pairs in there that are noisy. *Why isn't the solution to noisy image-text pairings just to have a lot more data?* That seems to be the answer CLIP provides and it is the state-of-the-art image-text retrieval model at present. Moreover, anybody can take CLIP and finetune it on a dataset if they feel there is too much domain shift.
The authors should have showcased a result showing why just having a ton of noisy image-text data isn't enough and that they can do better on less data by explicitly managing / modeling the noise. At present, I am not convinced I should reach for the authors' method over just reaching for a model trained on a large noisy dataset. This is a key point the authors should address.
Citation for CLIP:
Contrastive Language–Image Pre- training) (Radford et al., 2021)

Relationship with hard / soft negative mining methods, weighting approaches, and adaptive margin methods:
While the experimental comparison against a few recent retrieval methods is OK, the key point is to evaluate your experimental technique against other methods that do the same thing in different ways. For example, there are many papers that do a soft hard-negative mining technique (see Mithun, 2018 for the soft triplet loss), where they weigh each sample according to the distance. Similarly, there are papers that try to measure properties of the surrounding space and integrate it as a weight into the loss (e.g. Hubness aware loss). Others still try different mechanisms for learning an adaptive margin. I would have liked to see a comparison of why your weighting approach is better or at least a thorough discussion of these in related work.

Moreover, onto your approach, I understand that you use the confidence of the sample to compute the margin, but shouldn't we also consider the confidence of the negative samples as well? For example, if the confidence of the sample is very high then your method would say this sample should have a large margin. But if we are in a batch that has noisy negatives where the confidence is very low, then your method might try to enforce a large margin between the positive sample and the negatives - but those negatives could be noisy and actually shouldn't be that far from the positive. In other words, the margin should take into account the relationship of the positive sample as well as each negative to calculate the adaptive margin.

For the approach, I would also point out that the co-teaching technique used is really an adaptation of what has previously been done in the label space. Other prior work also uses a co-teaching approach to correct noisy data. So, the actual methodology of doing so isn't new, but the application to noisy image-text pairings and the way it is done by integrating it as a weight is new.

# Conclusion
The authors present a reasonable technique for correcting noisy image-text pairings. However, the scenario where the proposed method actually would apply has not been thoroughly explored. Moreover, if the authors wish to argue their approach does apply to scenarios where labels could be defined, they should have compared against noisy or self-supervised labeling methods. I believe the authors should also explain why the approach will not ignore the types of samples emphasized by the ECCV20 paper - i.e. the authors approach might learn to ignore the samples that have an indirect image-text relationship, at the expense of losing semantics from the representation space. Moreover, since they wish to argue web-scale data harvested from the web is often noisy, it isn't clear that the solution to that isn't just to use a lot more web-scale data. In that case, they should have compared against CLIP - which is the standard image-text retrieval method at the moment, is trained on noisy image-text pairings, and is available. In sum, while the approach attempts to solve an important and interesting problem, I am concerned about the evaluation scenarios and the lack of discussion of the issues mentioned.

# Final Comment (Post Rebuttal)
I thank the authors for their very thorough responses. I have reviewed the authors' results on CLIP and I encourage them to include these comparisons in their submission. Given that the performance is superior for some cases, and particularly because the performance is superior on the released model (CLIP) relative to the unreleased model, I would recommend the paper be accepted. I appreciate the authors' discussion of complementarity vs noise. This is an interesting issue that can be discussed in the text. Given the thorough results provided by the authors, I raise my rating to accept.

**Time Spent Reviewing:**

5

---

> ### Author Response · Authors · 2021-08-10
> **The Response to Reviewer 9fra (Part 1)**
>
> Thanks for your positive comments and insightful suggestions. Before getting into the detailed response, we summarize the **major concerns** from the reviewer as below for a better response.
>
> One major concern of the reviewer is about the application and evaluation scenarios of NCR. The reviewer pointed out that NCR is only applicable to the datasets with easily defined labels and explicit image-text relationships like Flickr30K and COCO, while failing to the datasets with unknown labels and indirect image-text relationships defined in [ref.A] due to an implicit assumption, namely, unclean pairs will have lower similarity after warmup while clean pairs will have a higher similarity. For the first scenario (easily defined labels), the reviewer suggests using a noisy or self-supervised labeling method as an alternative solution to noisy correspondence (**Q1**). For the second scenario, the reviewer expects an explanation of why NCR could deal with the complementary pairs instead of treating it as a kind of noisy correspondence,  as well as what makes NCR works well on the CC152K (Conceptual Captions) dataset which is with the complementary pairs  (**Q2**).
>
> Moreover, the reviewer emphasizes that NCR targets on the mismatching large-scale data harvested from the web, and suggests to compare with CLIP [B] which uses hundreds of million data to implicitly alleviate the noisy correspondence issue. Such a comparison is helpful in understanding, big data (CLIP) or noisy correspondence modeling technique (NCR), which one is more favorable (**Q3**).
>
> In the following, we will attempt to address these concerns one by one.
>
>
>
> **Q1**:  Alternatively using noisy or self-supervised labeling method in the scenario of easily defined labels.
>
> Thanks, it is not easy to solve the noisy correspondence problem using noisy and self-supervised methods. To be specific, the reviewer suggests, *to identify a set of objects in the captions, train a model to classify them, and then do some prediction of labels - and voila, you have a self-supervised and noisy set of labeled data*. Such a pipeline would fail unless the following challenges are well addressed. First, it is a daunting task to represent the semantic content of images only using the entities detected from the caption due to the difference in the information granularity of image and text. For example, it would fail in retrieving a complex scene with various objects. Second, the suggested approach consists of  three steps each of which cannot obtain perfect results, thus leading to error accumulation and further degrading the matching performance. Third, extracting object labels from the captions will suffer from the label ambiguity issue, i.e., the tags of the same object extracted from different captions might be different. Finally, even though the above challenges are well addressed, it is unclear how to use the trained classifier to perform matching.
>
>
>
> **Q2**: Why NCR could experimentally perform well on the CC152K dataset of which the image-text pairs are with indirect relationship, i.e., why NCR does not treat these complementary pairs as noisy correspondence?
>
> The thought of the reviewer is very interesting and inspiring, and we answer this question from the following aspects. First, the CC152K dataset contains not only the complementary pairs (indirect relationship) but also the pairs with explicit and noisy relationships, see some examples in Fig.4 of our submission and Figs.2--3 of the attached Supplemental Material. Second, we think there is **NO** absolute boundary among the three types of pairs (indirect, explicit, and noisy relationship) in spite of the practical scenario. In other words, the boundary is adaptively determined by the downstream tasks and the observed data. Taking the textual query "apple" as an example, it is unexpected to retrieve a picture of Newton if one does not hear the story of Newton's apple. Clearly, **additional** data or knowledge will transform these three types of relationships mutually. Third, as our method utilizes the memory effect of neural networks to distinct the clean pairs from noisy pairs, it could avoid treating the complementary pairs as noisy correspondence in a data-driven way. Considering our experiments, for the well-labeled data Flickr and COCO which consists of pairs with explicit relationship, the weakly related pairs like "apple" and picture of Newton should be considered as noisy samples (this is what NCR have done). While in the CC152K, NCR would assign high confidence to the explicitly and indirectly related pairs, and treat the completely unrelated pairs as noisy correspondence. Despite the above clarifications, we totally agree with the reviewer that it is promising and valuable to explore learning with complementary pairs, and the next version will include a more detailed discussion and analysis on the difference with related works and settings.
>
>
>
> **Q3**: Is explicit noisy correspondence modeling technique (NCR) superior to the pretrained model with big data (CLIP)? Requiring comparison to the CLIP pretrained on large-scale web data.
>
> As known, the controversy of **data vs. algorithm** is one major focus of the community. CLIP [ref.C] claims that using hundreds of million data could ignore the existence of possible noise, while we believe that a well-designed algorithm is essential to solve the noisy correspondence. It should be pointed out that, although existing works including CLIP and ALIGN [ref.G] has awarded the existence of noise, **NONE** of them explicitly explores the characteristics and types of the noises.
>
> As suggested, we conduct CLIP on the MS-COCO dataset under the following two settings, i.e., Zero-shot and Finetune. In brief, the first setting directly uses the released pretrained CLIP to perform inference on MS-COCO, and the second finetunes the pretrained model using the noisy training data of MS-COCO. As OpenAI only released the CLIP model (ResNet-50, ResNet-101, ViT-B/16, ViT-B/32) and inference code (https://github.com/OpenAI/CLIP), we use the non-official code from https://github.com/Zasder3/train-CLIP-FT to fine-tune the model with 32 epochs. In addition, we perform the retrieval task by using the code from https://github.com/openai/CLIP/issues/115. Due to the time limitation, we could only obtain the result of the two largest available models (CLIP-ResNet-101 and CLIP-ViT-B/32) as follows:
>
> |                       | Models                                                     | I2T         | I2T         | I2T         | T2I         | T2I         | T2I         |
> | --------------------- | ---------------------------------------------------------- | ----------- | ----------- | ----------- | ----------- | ----------- | ----------- |
> |                       |                                                            | R@1         | R@5         | R@10        | R@1         | R@5         | R@10        |
> | Zero-shot             | CLIP (ViT-L/14), **unreleased model**, reported in [ref.C] | 58.4        | 81.5        | 88.1        | 37.8        | 62.4        | 72.2        |
> | Zero-shot             | CLIP (ResNet-101)                                          | 49.2        | 73.9        | 82.5        | 30.7        | 55.3        | 66.0        |
> | Zero-shot             | CLIP (ViT-B/32)                                            | 50.2        | 74.6        | 83.6        | 30.4        | 56.0        | 66.8        |
> | 0% Noise              | NCR                                                        | 58.2        | 84.2        | 91.5        | 41.7        | 71.0        | 81.3        |
> |                       |                                                            |             |             |             |             |             |             |
> | Finetune on 20% Noise | CLIP (ResNet-101)                                          | In progress | In progress | In progress | In progress | In progress | In progress |
> | Finetune on 20% Noise | CLIP (ViT-B/32)                                            | 21.4        | 49.6        | 63.3        | 14.8        | 37.6        | 49.6        |
> | 20% Noise             | NCR                                                        | 56.9        | 83.6        | 91.0        | 40.6        | 69.8        | 80.1        |
> |                       |                                                            |             |             |             |             |             |             |
> | Finetune on 50% Noise | CLIP (ResNet-101)                                          | 20.9        | 42.6        | 54.5        | 12.7        | 30.3        | 40.4        |
> | Finetune on 50% Noise | CLIP (ViT-B/32)                                            | 10.9        | 27.8        | 38.3        | 7.8         | 19.5        | 26.8        |
> | 50% Noise             | NCR                                                        | 53.1        | 80.7        | 88.5        | 37.9        | 66.6        | 77.8        |
>
> One could observe that, although CLIP utilizes 400 million image-text pairs for pretraining, its performance inevitably degenerates during finetuning. In contrast, NCR achieves the matching performance with the presence of noisy correspondence, indicating the necessity of algorithm design.
>
> For the suggestion of extending our idea to CLIP, we would leave it in future works due to time limitation and unavailable large-scale training data.

---

> > ### Author Response · Authors · 2021-08-10
> > **The Response to Reviewer 9fra (Part 2)**
> >
> >
> >
> > **Q4**: Related works. Missing discussion to other soft triplet loss.
> >
> > Due to the time limitation, we could only conduct experiments for **Q3** and leave the analysis experiments in the future by integrating these possible losses into our framework. Moreover, the next version will add the discussions on the difference between these existing methods and NCR. In brief, [ref.D] recasts the soft similarity into binary labels by directly setting a threshold on the predicted similarity. [ref.E] proposes a log-ratio matching loss with a regularization defined by the label distance ratio, which is computed by the continuous labels. [ref.F] introduces the hubness problem in image-text matching and proposes to consider all samples in a mini-batch and weights them according to both local and global statistics. Different from them, NCR proposes to recast the rectified soft labels into soft margins in the triplet loss by assigning large margins to the true positive pairs and small ones to the false positive pairs. Moreover, another major difference between our loss and these soft triplet losses is that the former is specifically designed to solve the noisy correspondence problem whereas the latter did not.
> >
> >
> >
> > **Q5**: Why does the adaptive margin only consider the confidence of positive pairs while ignoring the negatives?
> >
> > The negatives are unnecessary to compute the adaptive margins in our method. We assume that the reviewer might misunderstand the role of $L_{soft}$ in Eq. 6. Specifically, $L_{soft}$ aims to enforce the given positive pairs are closer than the hard negatives in a mini-batch with an adaptive margin (see Fig.2). The adaptive margins are computed using the rectified labels defined in Eq. 7. For the pointed outed case that the true positives are incorrectly rectified to noisy one ($y\to0$), $L_{soft}$ will still enforce them closer than the hard negatives with a small margin. Hence, it will not happen that the noisy negatives (i.e., positive) are pushed away from the positives.
> >
> >
> >
> > **Q6**: Methodology novelty.
> >
> > We agree that co-teaching has been well developed in recent years in noisy label learning. However, we would highlight that this is the first work on noisy correspondence, a new research direction of the community. As stated in our manuscript, we think noisy correspondence could be a new paradigm of the noisy label, which motivates us to solve it by referring to some existing techniques. Despite the novelty and significance of the problem, NCR is not a direct application of co-teaching. In brief, we elegantly reformulate the rectiﬁed label as the soft margin of a triplet loss so that the robustness is achieved, and such a reformulation is nontrivial.
> >
> >
> >
> > **References:**
> >
> > - [A] Thomas, Christopher, and Adriana Kovashka. "Preserving semantic neighborhoods for robust cross-modal retrieval." European Conference on Computer Vision. Springer, Cham, 2020.
> > - [B] Radford, A., Kim, J. W., Hallacy, C., Ramesh, A., Goh, G., Agarwal, S., ... & Sutskever, I. (2021). Learning transferable visual models from natural language supervision. *arXiv preprint arXiv:2103.00020*.
> > - [C] Sharma, P., Ding, N., Goodman, S., & Soricut, R. (2018, July). Conceptual captions: A cleaned, hypernymed, image alt-text dataset for automatic image captioning. In *Proceedings of the 56th Annual Meeting of the Association for Computational Linguistics (Volume 1: Long Papers)* (pp. 2556-2565).
> > - [D] Wray, Michael, Hazel Doughty, and Dima Damen. "On Semantic Similarity in Video Retrieval." Proceedings of the IEEE/CVF Conference on Computer Vision and Pattern Recognition. 2021.
> > - [E] Kim, Sungyeon, et al. "Deep metric learning beyond binary supervision." Proceedings of the IEEE/CVF Conference on Computer Vision and Pattern Recognition. 2019.
> > - [F] Liu, F., Ye, R., Wang, X., & Li, S. (2020, April). Hal: Improved text-image matching by mitigating visual semantic hubs. In *Proceedings of the AAAI Conference on Artificial Intelligence*(Vol. 34, No. 07, pp. 11563-11571).
> > - [G] Jia, C., Yang, Y., Xia, Y., Chen, Y. T., Parekh, Z., Pham, H., ... & Duerig, T. (2021). Scaling up visual and vision-language representation learning with noisy text supervision. *arXiv preprint arXiv:2102.05918*.

---

> ### Author Response · Authors · 2021-08-24
> **Response to the Final Comment**
>
> Thanks for your positive comments and suggestions. We would include the comparison experiment to CLIP and the discussion with the referred works in the next version as suggested.

---

### Decision · Program_Chairs · 2021-09-27

**Decision:**

Accept (Oral)

**Comment:**

The paper initially received positive reviews from four reviewers (i.e., 6 7 7 8). In the rebuttal, the authors address the concerns from the reviewers very well (e.g., comparison with CLIP), and then all reviewers further raise their scores (to 7 8 9 9).

After carefully reading the manuscript, comments, and the corresponding response, I agree with the reviewers that this paper makes a significant contribution to the community, namely, revealing a NEW problem (i.e., noisy correspondence/mismatching pairs) in practice for the first time. The work is certainly valuable to the community including but not limited to cross-modal matching because the essence of many tasks such as ReId, visual grounding, and VQA, is exploring the correspondence of paired data.

Overall, this paper is with strong motivation, a technical sound approach, extensive experiments, and good writing.